# isoCirc catalogs full-length circular RNA isoforms in human transcriptomes

Ruijiao Xin[1,5], Yan Gao [1,2,5], Yuan Gao[1], Robert Wang[3], Kathryn E. Kadash-Edmondson [1], Bo Liu[2], Yadong Wang[2], Lan Lin[4] & Yi Xing [1,4 ✉]

Circular RNAs (circRNAs) have emerged as an important class of functional RNA molecules. Short-read RNA sequencing (RNA-seq) is a widely used strategy to identify circRNAs. However, an inherent limitation of short-read RNA-seq is that it does not experimentally determine the full-length sequences and exact exonic compositions of circRNAs. Here, we report isoCirc, a strategy for sequencing full-length circRNA isoforms, using rolling circle amplification followed by nanopore long-read sequencing. We describe an integrated computational pipeline to reliably characterize full-length circRNA isoforms using isoCirc data. Using isoCirc, we generate a comprehensive catalog of 107,147 full-length circRNA isoforms across 12 human tissues and one human cell line (HEK293), including 40,628 isoforms ≥500 nt in length. We identify widespread alternative splicing events within the internal part of circRNAs, including 720 retained intron events corresponding to a class of exon-intron circRNAs (ElciRNAs). Collectively, isoCirc and the companion dataset provide a useful strategy and resource for studying circRNAs in human transcriptomes.

[1] Center for Computational and Genomic Medicine, The Children's Hospital of Philadelphia, Philadelphia, PA 19104, USA. [2] Department of Computer Science and Technology, Center for Bioinformatics, Harbin Institute of Technology, Harbin, Heilongjiang 150001, China. [3] Genomics and Computational Biology Graduate Program, University of Pennsylvania, Philadelphia, PA 19104, USA. [4] Department of Pathology and Laboratory Medicine, University of Pennsylvania, Philadelphia, PA 19104, USA. [5] These authors contributed equally: Ruijiao Xin, Yan Gao. ✉email: xingyi@email.chop.edu

Circular RNAs (circRNAs) are endogenous RNA molecules with a continuous loop structure, produced by back-splicing events between a downstream 5′ splice site and an upstream 3′ splice site[1–5]. CircRNAs may contain exonic or intronic sequences of their parental genes and range in size from very small (100 nt) to more than 4 kb[1]. They have been implicated in multiple molecular processes, such as regulation of transcription[6], regulation of microRNA or protein binding to RNA targets[7–9], and translation into protein products[10–12]. Nevertheless, despite growing interest in circRNAs, the functional roles of most circRNAs remain unknown.

In the last decade, short-read RNA sequencing (RNA-seq) has become an effective and widely used strategy to characterize circRNAs. By identifying and counting sequence reads that map to back-splice junctions (BSJs), circRNAs can be discovered and quantified using short-read RNA-seq data[1,13]. In a pioneering study, by analyzing RNA-seq data of multiple human cell types, Salzman and colleagues provided the first evidence for the widespread occurrence of circRNA BSJs in the human transcriptome. They also observed that circRNAs are the predominant transcript isoform in hundreds of human genes[14]. This finding was corroborated by subsequent RNA-seq studies showing that circRNAs are abundant and conserved in animals[8,15]. Numerous computational tools have been developed to accurately discover and quantify circRNA BSJs from short-read RNA-seq data (reviewed in refs. [1,13]). Using short-read RNA-seq, researchers have observed widespread changes in circRNA expression profiles among different tissues and cell types[16–20], in response to cellular signals[21], and during disease pathogenesis[22–24].

Despite the tremendous successes of short-read RNA-seq studies of circRNAs, an inherent limitation of this approach is that short-read RNA-seq does not experimentally determine the full-length sequences and internal alternative splicing events within circRNAs[13]. Very few circRNAs have been functionally characterized, and functional studies of circRNAs substantially benefit from knowledge of full-length circRNA sequences[13]. For example, to identify circRNAs that act as sponges for microRNAs or RNA-binding proteins, it is essential to know their full-length sequences (as opposed to the BSJs alone)[9]. Similarly, inferring the protein products translated from circRNAs requires full-length circRNA sequences[11]. To fill this gap, several computational methods have been developed to reconstruct full-length circRNAs from short-read RNA-seq data[25–29]. However, these methods are only applicable to short circRNAs (200–500 nt, depending on RNA-seq read length), and are unable to interrogate longer full-length circRNAs[25].

Long-read sequencing has recently emerged as a promising and versatile technology for transcriptome analysis[30–32]. Long-read sequencing platforms such as Pacific Biosciences (PacBio) and Oxford Nanopore Technologies (ONT) can generate reads that are thousands to tens of thousands of bases in length, making long-read RNA-seq a powerful tool for resolving full-length transcript isoforms. In this manuscript, we report isoCirc, a strategy for sequencing full-length circRNA isoforms, using rolling circle amplification (RCA) followed by nanopore long-read sequencing. We describe an integrated computational pipeline to reliably characterize full-length circRNA isoforms using isoCirc data. Using isoCirc, we generated a comprehensive catalog of full-length circRNA isoforms across 12 human tissues and one human cell line (HEK293). We identified widespread alternative splicing events within the internal part of circRNAs, including retained intron events corresponding to a class of exon–intron circRNAs (EIciRNAs).

## Results

**Experimental and computational workflow of isoCirc.** We have developed isoCirc, a strategy that combines RCA and nanopore long-read sequencing to characterize full-length circRNA isoforms. The isoCirc workflow is summarized in Fig. 1, and details of the experimental and computational procedures are described in Methods. Briefly, from total RNAs extracted from a biological sample, circRNAs are enriched and linear RNAs are depleted through ribosomal RNA (rRNA) removal and RNase R treatment. A random primer is used to initiate reverse transcription (RT) of the circRNA template. If the reverse transcriptase processes the circRNA beyond a full circle, then the 5′ overhang of the RT product is removed by nuclease digestion. The RT product is ligated into a circular cDNA, which is then amplified through RCA. The final RCA product is subjected to long-read sequencing on an ONT sequencer. Resulting raw reads are expected to contain multiple copies of a template sequence that initiates from a random position within the circRNA during the RT reaction. To identify full-length circRNAs that match this expected pattern, tandem repeats are detected from isoCirc raw reads and used to generate consensus sequences. A concatemer of two copies of the consensus sequence is mapped to the genome to identify the BSJ and forward-splice junctions (FSJs) within a circRNA. Alignment results are examined for multiple stringent criteria (e.g., mapping quality, fidelity in the putative BSJs and FSJs; see details in Supplementary Methods) to identify high-confidence BSJs and full-length circRNA isoforms.

**isoCirc analysis of the HEK293 cell line.** As an initial proof-of-concept study, we applied the isoCirc experimental workflow to RNAs from the HEK293 cell line. We generated nanopore long-read sequencing data for six HEK293 libraries (two biological replicates, each with three technical replicates). We generated 20.3 million reads, with 2.6 to 4.2 million reads per library (Supplementary Table 1). Applying the isoCirc computational pipeline to the data, we detected between 47,434 and 66,194 high-confidence BSJs and between 41,095 and 57,763 full-length circRNA isoforms per library.

We characterized the features of isoCirc raw reads and consensus sequences. The isoCirc raw reads corresponded to RCA products of circRNA isoforms, each containing multiple copies of a template circRNA sequence. Therefore, the consensus sequences called from isoCirc raw reads are expected to have a reduced error rate. To confirm this, for all isoCirc reads for which consensus sequences were called and mappable to the human genome, we calculated the error rate and length of both the raw reads and the consensus sequences. As expected, the raw reads had the highest error rate. We observed a copy number-dependent decrease in the error rate of consensus sequences, with consensus sequences called from raw reads with more than 10 copies having the lowest error rate (Supplementary Fig. 1). Among the isoCirc raw reads and consensus sequences that met our stringent criteria to support high-confidence BSJs, the average copy number of consensus sequences within raw reads was 14.5, suggesting that high-accuracy consensus sequences were used to identify circRNA BSJs and full-length isoforms. Specifically, the average length of raw reads was 5679 nt, whereas the average length of consensus sequences was 460 nt. Length distributions of the raw reads and consensus sequences were comparable among the six HEK293 replicates (Supplementary Fig. 2), supporting the technical reproducibility of the isoCirc experimental workflow.

We performed a set of analyses to evaluate the reproducibility of the isoCirc results at the BSJ and isoform levels. First, for a given pair of two libraries, we calculated their degree of similarity in identified BSJs or full-length circRNA isoforms by dividing the

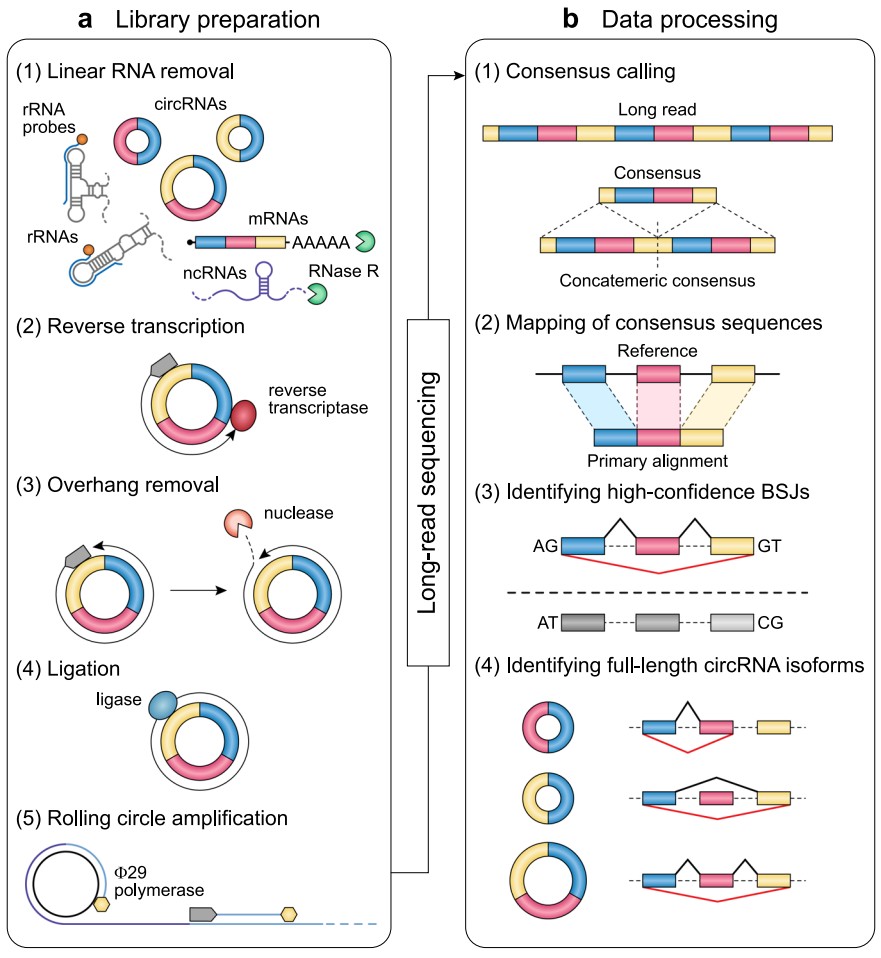

**a** Library preparation

(1) Linear RNA removal
rRNA probes
circRNAs
rRNAs
mRNAs
AAAAA
ncRNAs
RNase R

(2) Reverse transcription
reverse transcriptase

(3) Overhang removal
nuclease

(4) Ligation
ligase

(5) Rolling circle amplification
Φ29 polymerase

Long-read sequencing

**b** Data processing

(1) Consensus calling
Long read
Consensus
Concatemeric consensus

(2) Mapping of consensus sequences
Reference
Primary alignment

(3) Identifying high-confidence BSJs
AG — GT
AT — CG

(4) Identifying full-length circRNA isoforms

**Fig. 1 Overall experimental and computational workflow of isoCirc. a** Preparation of circular RNA (circRNA) libraries for isoCirc. Extracted total RNAs are subjected to linear RNA removal via ribosomal RNA (rRNA) depletion and RNase R treatment to enrich for circRNAs. The circRNA template is reverse transcribed and digested with nuclease to remove any 5′ overhang of the reverse transcription (RT) product. Ligation of the RT product generates the circular cDNA template. Rolling circle amplification of the circular cDNA template generates the sample for long-read sequencing. ncRNA: noncoding RNA. **b** Processing of long-read isoCirc sequencing data. In the consensus calling step, tandem repeats are detected from long reads and used to generate consensus sequences. For each read, a concatemer of two copies of the consensus sequence is mapped to the genome to identify the back-splice junction (BSJ) and forward-splice junctions (FSJs) within the circRNA. Alignment records are filtered using multiple stringent criteria (e.g., mapping quality, BSJ/FSJ fidelity). In this way, isoCirc enables identification of high-confidence BSJs and full-length circRNA isoforms.

number of BSJs or isoforms detected in both libraries by the total number of unique BSJs or isoforms detected in either library. Applying this metric to the six HEK293 libraries, we found agreement among isoCirc replicates at both the BSJ level (Fig. 2a) and the isoform level (Fig. 2b). Degree of similarity among replicates was higher when we required that identified BSJs (Supplementary Fig. 3) or full-length isoforms (Supplementary Fig. 4) be supported by a larger number of isoCirc reads. As expected, technical replicates from the same biological replicate had slightly higher similarity than technical replicates from different biological replicates (Fig. 2a, b). This pattern was consistent across different read-count cutoffs for BSJs or full-length isoforms (Supplementary Figs. 3, 4). Second, we compared isoCirc read counts for BSJs (Supplementary Fig. 5) and full-length isoforms (Supplementary Fig. 6) among the six replicates. We found that isoCirc quantitation of BSJs and isoforms agreed well among the six replicates, especially for abundantly expressed circRNAs. In this work, "read count" refers to the number of independent nanopore reads supporting a given circRNA BSJ or full-length isoform. The isoCirc read count is distinct from the copy number of a given template circRNA sequence within a nanopore read (i.e., RCA product).

**isoCirc versus short-read RNA-seq analysis of circRNA BSJs.** Next, we compared isoCirc to short-read RNA-seq analyses of circRNA BSJs. We generated 189.4 million and 212.6 million Illumina RNA-seq reads (101 bp × 2) on three RNase R-treated and three poly(A)-selected libraries of HEK293 cells, respectively (Supplementary Table 2).

First, to assess the ability of the isoCirc experimental workflow to enrich circRNAs, we compared the proportion of reads that mapped to circRNA BSJs in the isoCirc datasets versus RNase R-treated or poly(A)-selected Illumina RNA-seq datasets of HEK293 cells (see Methods). The proportion of circRNA BSJ reads in isoCirc libraries ranged between 3.5 and 4.0%. By comparison, the proportion of circRNA BSJ reads in Illumina libraries was ~0.001% for poly(A)-selected libraries and ranged between 0.045 and 0.048% for RNase R-treated libraries (Supplementary Fig. 7, Supplementary Table 2). Thus, we observed an over 70-fold enrichment of circRNA BSJ reads in isoCirc libraries compared to RNase R-treated Illumina libraries. Even considering that not all reads derived from circRNAs in Illumina short-read libraries map to circRNA BSJs, these results still illustrate the ability of the isoCirc experimental workflow to substantially enrich for circRNAs.

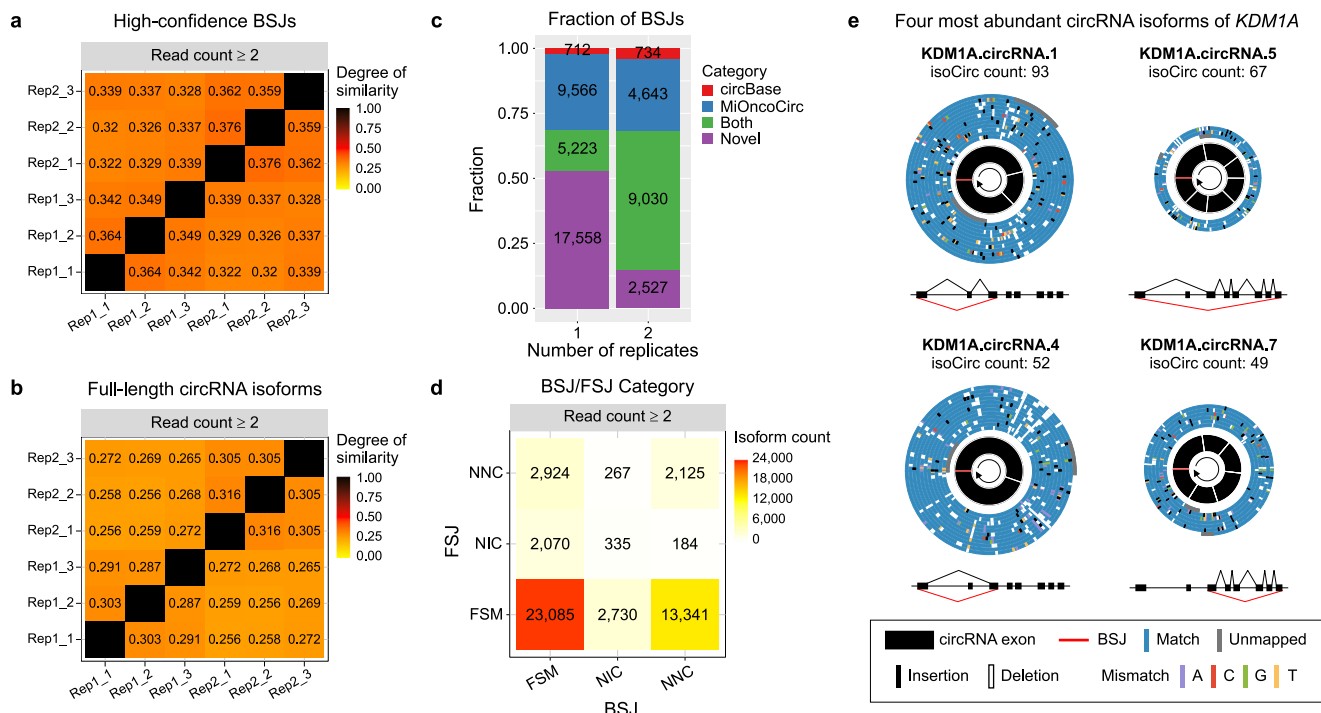

**Fig. 2 isoCirc data of HEK293 cells. a** Heatmap showing pairwise comparison of similarity between high-confidence BSJs identified from six HEK293 libraries. Only BSJs with read count ≥2 were included. For each library pair, degree of similarity was calculated as number of shared high-confidence BSJs found in both libraries divided by total number of high-confidence BSJs in either library. Color reflects degree of similarity between two libraries, as indicated in legend. **b** Heatmap showing pairwise comparison of similarity between full-length circRNA isoforms (with read count ≥2) identified from six HEK293 libraries. Degree of similarity was determined as in (**a**). **c** Stacked barplot showing fraction of known or novel high-confidence BSJs identified in only one ('1') or both ('2') biological replicates of HEK293 cells, based on BSJ annotations in circBase (http://www.circbase.org) and MiOncoCirc (https://mioncocirc.github.io) databases. Only BSJs with read count ≥2 in each biological replicate (summing over three technical replicates) were included. Bars show known BSJs annotated in circBase only (red), MiOncoCirc only (blue), or both databases ('Both', green), and novel BSJs not annotated in either database ('Novel', purple). **d** Heatmap showing numbers of full-length circRNA isoforms identified in HEK293 cells, based on BSJ (x-axis) and FSJ (y-axis) categories as classified relative to existing transcript annotations. Only full-length circRNA isoforms with read count ≥2 (summing over all six libraries) were included. All identified circRNA BSJs and FSJs were classified using three categories: Full Splice Match (FSM), Novel In Catalog (NIC), and Novel Not in Catalog (NNC). **e** Long-read alignments (top) and gene structure diagrams (bottom) for the four most abundant full-length circRNA isoforms of *KDM1A* in HEK293 cells, as measured by isoCirc read count. All four isoforms were categorized as FSM for both BSJ and FSJs. Long-read alignments indicate multiple copies of circRNA templates in isoCirc reads. Inner black circle: circRNA gene structure with BSJ (red line) and FSJs (white lines). Blue circles: matched bases of isoCirc sequences aligned to reference genome sequence. Colored lines: mismatched bases (purple: A, red: C, green: G, yellow: T), insertions (black), and deletions (white), compared to reference genome sequence.

Second, to assess the reproducibility of short-read RNA-seq versus isoCirc among replicates, we used BSJs detected from three RNase R-treated Illumina RNA-seq libraries to assess the degree of similarity among the three short-read replicates (Supplementary Fig. 8). Comparing this result to the degree of similarity among isoCirc technical replicates (Fig. 2a and Supplementary Fig. 3), we observed that isoCirc had comparable (read count ≥2) to slightly higher (read count ≥3) reproducibility, as compared to Illumina RNA-seq of RNase R-treated samples at the same read-count threshold.

Third, to compare BSJ quantitation among isoCirc and short-read RNA-seq, we compared BSJ counts from isoCirc to short-read (RNase R-treated) BSJ counts combining all replicates. At read count ≥2, we observed a Spearman correlation of 0.5 between the isoCirc data and the short-read data (Supplementary Fig. 9), indicating a reasonable agreement between these two distinct methods for BSJ quantitation.

**isoCirc discovery of known versus novel circRNA isoforms**. We next assessed the ability of isoCirc to identify full-length circRNA isoforms with known or novel splicing events. As an initial analysis, we combined the three technical replicates of each

HEK293 biological replicate, and compared identified BSJs in the two biological replicates with known circRNA BSJs in the circBase[33] and MiOncoCirc[22] databases (Fig. 2c, Supplementary Table 1). At read count ≥2, we identified 49,993 high-confidence BSJs. Of the 33,059 high-confidence BSJs found in only one of the two HEK293 biological replicates ('1' in Fig. 2c), 46.9% were known BSJs documented in circBase[33] and/or MiOncoCirc[22], and 53.1% were novel BSJs found in neither database. Of the 16,934 BSJs found in both biological replicates ('2' in Fig. 2c), a much higher percentage (85.1%) were known, while a nontrivial percentage (14.9%) were novel. We observed the same trend at read count ≥3, although a higher proportion of identified BSJs at this read-count cutoff were known BSJs (Supplementary Fig. 10). In addition to validating the ability of isoCirc to detect known circRNA BSJs, these results demonstrate that isoCirc can identify novel BSJs that are not cataloged in existing circRNA databases.

We compared the features of known versus novel BSJs identified by isoCirc. As expected, known BSJs had higher read counts than novel BSJs, although the difference was modest, especially between known and novel BSJs detected in only one biological replicate (Supplementary Fig. 11). We further investigated whether known or novel BSJs identified by isoCirc were enriched for inverted Alu repeats in flanking introns, a feature

associated with the biogenesis of some (but not all) circRNAs[34]. Specifically, for each BSJ, we checked two windows (1000 and 2000 nt) of the genomic sequence flanking the back-splice sites. For each window, we asked whether the window contained inverted Alu repeats in the convergent orientation, divergent orientation, both orientations, or did not contain any inverted Alu repeats. We analyzed four sets of BSJs (known or novel BSJs identified in only one or both biological replicates), plus a fifth set of negative-control BSJs (obtained by making 10,000 random pairs of a downstream 5′ splice site and an upstream 3′ splice site, using non-BSJ splice sites based on isoCirc results and circRNA databases). Both known and novel BSJs (versus negative-control BSJs) were significantly enriched for inverted Alu repeats ($p$-value < 3.6e-50 in all comparisons, Fisher's exact test). As expected, a higher proportion of known BSJs (versus novel BSJs) had inverted Alu repeats in their flanking introns, and a higher proportion of BSJs identified in both biological replicates (versus BSJs in only one replicate) had inverted Alu repeats (Supplementary Fig. 12). Therefore, a higher proportion of novel BSJs lacked the characteristic feature of inverted Alu repeats in flanking introns for their biogenesis.

To systematically characterize the structure (BSJ and FSJs) of full-length circRNA isoforms identified by isoCirc, we adopted a classification system from long-read RNA-seq analysis of linear RNA transcripts[35]. We classified the BSJ or FSJs of each circRNA isoform into three categories: Full Splice Match (FSM), Novel In Catalog (NIC), and Novel Not in Catalog (NNC) (see Methods for details). Intuitively, FSM indicates that the full combination of splice junctions matches existing annotations of known circRNA BSJs or linear RNA transcripts; NIC involves novel combinations (splice junctions) of known splice sites in existing gene annotation databases; and NNC involves novel splice sites not cataloged in existing gene annotation databases.

After combining all six replicates and requiring read count ≥2, we identified 47,061 full-length circRNA isoforms in the HEK293 cell line. We observed all nine possible combinations of classifications for the BSJ and FSJs (Fig. 2d). The largest category of circRNA isoforms (23,085) were classified as FSM for both the BSJ and FSJs. We also found 2070 or 2924 isoforms as FSM for the BSJ while NIC or NNC for the FSJs, respectively, indicating that the internal part of the circRNA incorporated novel forward-splice junctions of known (NIC) or novel (NNC) splice sites. Conversely, we identified 2730 and 13,341 isoforms for which the internal part (FSJs) matched known transcripts while the BSJ involved novel back-splice junctions of known (NIC) or novel (NNC) splice sites. Requiring a higher read-count cutoff for circRNA isoforms (≥3) reduced numbers in all nine categories, but a significant fraction of isoforms still incorporated novel BSJs or FSJs or both (Supplementary Fig. 13).

As an example of the isoCirc data, Fig. 2e shows four representative isoCirc reads of *KDM1A*, a member of the histone demethylase (KDM) family. At read count ≥2, we identified 20 distinct full-length circRNA isoforms, including 10 that were FSM for both the BSJ and FSJs. The four most abundant circRNA isoforms (as measured by isoCirc read count) in HEK293 cells contained 3, 7, 2, and 6 exons, respectively. The isoCirc reads shown in Fig. 2e contained multiple RCA copies of these four circRNA isoforms. Two isoforms (KDM1A.circRNA.1 and KDM1A.circRNA.4) utilized an identical BSJ but had alternative inclusion versus skipping of an internal exon between them. The other two isoforms used distinct BSJs. All four isoforms were FSM for both the BSJ and FSJs, and the BSJs were annotated in circBase[33] and MiOncoCirc[22]. Transcript structures and supporting isoCirc consensus sequences for these four *KDM1A* circRNA isoforms are illustrated in Supplementary Fig. 14.

**isoCirc analysis of 12 human tissues**. Next, we applied isoCirc to discover and quantify circRNA isoforms from 12 human tissues representing diverse anatomical sites. We generated a total of 110.7 M nanopore reads, with between 6.2 M and 17.0 M reads per tissue (Supplementary Table 1). Despite having comparable sequencing depths, the number of circRNA isoforms identified varied substantially across different tissues. For example, considering unique isoforms with read count ≥1, we identified 115,348 circRNA isoforms in testis, compared to 18,312 in prostate and 19,261 in skeletal muscle (Fig. 3a, Supplementary Table 1). To assess whether this variability reflects biological differences in circRNA complexity among tissues, or technical variability in isoCirc experiments, we compared isoCirc discovery of circRNAs to short-read discovery of circRNAs across tissues (see Methods for details). Specifically, we compared the read depth-normalized number of circRNA BSJs detected by isoCirc and by short-read RNA-seq in a published study[17], across a common set of eight human tissues (Fig. 3b, Supplementary Table 3). Strikingly, despite approximating circRNA complexity using metrics based on data from distinct technologies, these two measurements were highly correlated across the eight tissues (Fig. 3b; Pearson correlation coefficient of 0.86). Tissues with the highest numbers of circRNA BSJs/isoforms detected by isoCirc (brain, testis) also had the highest numbers of circRNA BSJs detected by short-read RNA-seq. Conversely, tissues with low numbers of circRNA BSJs/isoforms by isoCirc also had low numbers of circRNA BSJs by short-read RNA-seq. Consistent with the increased number of circRNAs in testis, brain, and blood, these three tissues also had higher isoCirc read counts supporting circRNA isoforms containing novel BSJ or FSJs (Supplementary Fig. 15).

We systematically characterized the diversity and features of full-length circRNA isoforms identified by isoCirc in the 12 tissues. Combining all 12 tissues and requiring read count ≥2 in at least one tissue, we observed all nine possible combinations of classifications (FSM/NIC/NNC) for the BSJ and FSJs (Supplementary Fig. 16). The largest category of circRNA isoforms (35,913 isoforms; 45.1%) was classified as FSM for both the BSJ and FSJs. The remaining isoforms contained a novel (NIC or NNC) BSJ and/or FSJs.

We also used the 12-tissue isoCirc data to evaluate the exon number, transcript length, and number of circRNA isoforms per gene. The median number of circRNA isoforms was 3 per gene (Supplementary Fig. 17a). This result appeared to derive primarily from distinct BSJs within genes because the majority (88.2%) of BSJs had only one isoform (Supplementary Fig. 17b). Except for isoforms classified as NNC for the BSJ and FSM or NIC for FSJs, which were enriched for single-exon circRNAs, other categories had similar distributions of exon number (Fig. 3c) and transcript length (Fig. 3d). Focusing on isoforms classified as FSM or NIC for both the BSJ and FSJs, we found that their median exon number was 3 and median transcript length was 426 nt. Of these FSM/NIC-FSM/NIC isoforms, 41.7% and 8.7% were ≥500 nt and 1000 nt, respectively, with the longest isoform being 2039 nt. Such long circRNA isoforms cannot be interrogated by short-read based reconstruction methods[25], illustrating a key advantage of long-read circRNA sequencing by isoCirc.

A useful feature of the isoCirc dataset is that it allows us to quantify and compare the circRNA isoform composition of any gene across 12 human tissues. For genes containing at least two circRNA isoforms, we performed all possible pairwise tissue comparisons using isoform-specific read counts (see Methods). We identified 1) genes with significant shifts in isoform composition (relative proportion of individual isoforms) across human tissues (Fig. 3e), and 2) subsequently the specific isoform(s) responsible

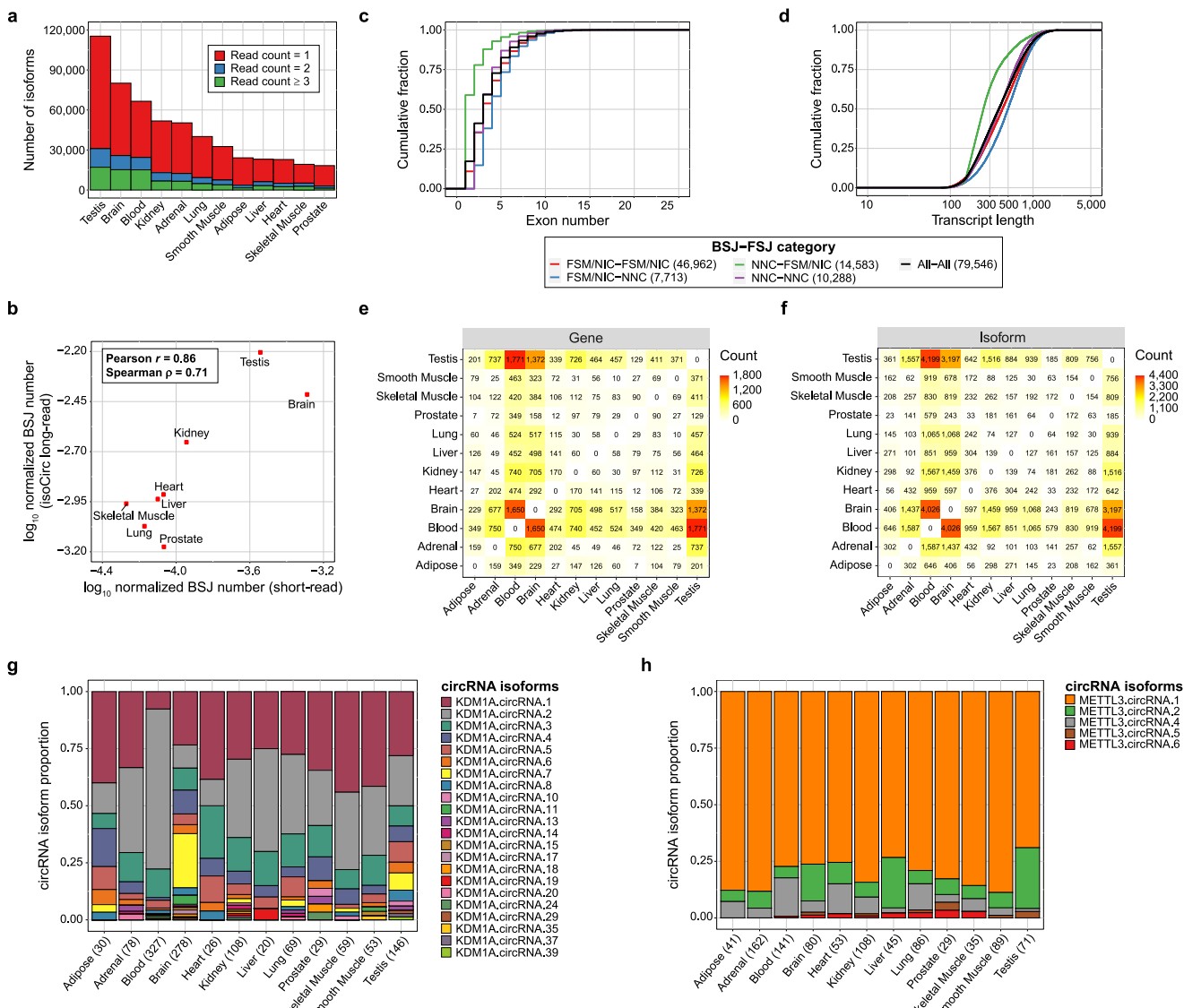

**Fig. 3 isoCirc characterization of circRNAs in 12 human tissues. a** Barplot showing number of full-length circRNA isoforms (*y*-axis) identified in each of 12 human tissues (*x*-axis), for read count = 1 (red), read count = 2 (blue), or read count ≥3 (green). **b** Correlation of number of circRNA BSJs identified from eight human tissues in published short-read datasets[17] and isoCirc long-read datasets. For each tissue, number of unique circRNA BSJs identified in either short-read (*x*-axis) or long-read (*y*-axis) data was normalized by the total number of short reads or long reads in that tissue. Data were plotted as log₁₀ transformed for ease of comparison and visualization. **c** Cumulative distribution plot of exon number per isoform for full-length circRNA isoforms with read count ≥2 in at least one of 12 human tissues. Isoforms were classified by their BSJ-FSJ categories, as follows: both BSJ and FSJs were FSM or NIC (red: FSM/NIC-FSM/NIC); BSJ was FSM or NIC, FSJs were NNC (blue: FSM/NIC-NNC); BSJ was NNC, FSJs were FSM or NIC (green: NNC-FSM/NIC); both BSJ and FSJs were NNC (purple: NNC–NNC); and all isoforms were combined (black: All–All). **d** Cumulative distribution plot of transcript length (nt) for full-length circRNA isoforms with read count ≥2 in at least one of 12 human tissues. Isoforms were classified by their BSJ-FSJ categories, as described in (**c**). **e** Heatmap showing numbers of genes with differential proportions of circRNA isoforms between each pair of 12 human tissues. **f** Heatmap showing numbers of circRNA isoforms with differential isoform proportions between each pair of 12 human tissues. **g** Stacked barplot showing isoform proportions of *KDM1A* circRNA isoforms across 12 human tissues. CircRNA isoforms were included in plot if read count was ≥2 in at least one tissue, and both BSJ and FSJs were FSM or NIC (FSM/NIC-FSM/NIC). Total read count for all circRNA isoforms in a tissue is given in parentheses on *x*-axis. **h** Stacked barplot showing isoform proportions of *METTL3* circRNA isoforms across 12 human tissues. Details are as in (**g**).

for the detected shifts in isoform composition at the gene level (Fig. 3f; Supplementary Fig. 18). Between different tissue pairs, we identified as few as 7 genes and 23 isoforms, and as many as 1771 genes and 4199 isoforms, that showed significant shifts in isoform composition across tissues.

For example, we detected a significant shift in circRNA isoform composition of *KDM1A* across tissues. One isoform (KDM1A. circRNA.7) constituted 23.5% of all isoforms in brain, as compared to 6.1% of all isoforms in testis and even lower

proportions in all other tissues (Fig. 3g). Other genes exhibited stable isoform compositions across tissues. *METTL3*, for instance, had one isoform (METTL3.circRNA.1) that was the predominant isoform across all tissues (Fig. 3h). Additional examples of circRNA isoforms showing tissue-specific or tissue-stable isoform composition across the 12 tissues are shown in Supplementary Figs. 19 and 20, respectively. In total, we identified 3653 tissue-specific circRNA isoforms with either tissue-specific expression or tissue-specific shifts in isoform proportion among the 12 tissues.

By checking the tissue specificity of gene expression patterns of their parental genes (see Methods), we found that tissue-specific circRNA isoforms were significantly enriched for tissue-specific genes as compared to random expectation (p-value < 2.2e-16, Fisher's exact test). As expected, this enrichment was attributed to tissue-specific genes in which all circRNA isoforms were exclusively detected in a single tissue.

**Alternative splicing of circRNAs revealed by isoCirc.** The iso-Circ data allowed us to investigate alternative splicing events of circRNAs, including internal alternative splicing events within circRNAs far from the BSJs. For genes with multiple circRNA isoforms identified by isoCirc, we compared the predominant isoform (i.e., with the highest median read count across the 12 tissues and HEK293 cell line) to each of the other isoforms in the gene. Of the 82,602 isoform pairs compared, 4.7% (3920), 7.2% (5913), and 88.1% (72,769) had alternative splicing differences in the BSJ only, FSJs only, or both (Fig. 4a). Focusing on the internal part of circRNAs, we identified 5325 alternative splicing events corresponding to four major types of alternative splicing patterns (skipped exon, alternative 5′ splice site, alternative 3′ splice site, retained intron), when requiring that the minor isoform in comparison had ≥2 isoCirc reads in all samples combined (Fig. 4b). At higher minor isoform read-count cutoffs of ≥5 and ≥10, we identified 2683 and 1310 internal alternative splicing events, respectively. Furthermore, for 2098 internal alternative splicing events, the splicing events of both isoforms matched known transcripts or utilized known splice sites (annotated as FSM/NIC; see Fig. 4b).

To investigate whether internal alternative splicing events discovered by isoCirc had unique features, we compared isoCirc-based results of circRNA internal alternative splicing to results on the HEK293 cell line from a previous short-read RNA-seq study[26]. In both studies, skipped exon was the most common type of internal alternative splicing patterns identified. Relative occurrences of skipped exons, alternative 5′ splice sites, and alternative 3′ splice sites were comparable between the short-read study and isoCirc, with isoCirc obtaining several (2.4–4.3) times more events. A notable exception was with retained introns, for which isoCirc identified close to 20 times more events compared to the short-read study[26]. In other words, compared to the short-read study, isoCirc identified a disproportionately higher number of retained introns within the internal part of circRNAs (Fig. 4b).

CircRNAs containing retained introns, termed as exon–intron circRNAs (EIciRNAs), were previously shown to regulate gene expression in the nucleus[6]. However, EIciRNAs are difficult to identify by short-read based methods, due to the large size of human introns[6]. Using isoCirc, we identified 720 retained intron events corresponding to EIciRNAs. Even when we required that the minor isoform had ≥10 isoCirc reads in all samples combined, we still obtained 177 events. Thus, isoCirc substantially expands the catalog of EIciRNAs in the human transcriptome. In one gene (PRPSAP1), the two most abundant circRNA isoforms (PRPSAP1.circRNA.1 and PRPSAP1.circRNA.2) had an alternative splicing event corresponding to a retained or spliced intron, with the predominant isoform defined based on the aggregated 12 tissue+HEK293 data (PRPSAP1.circRNA.1) containing the retained intron. This retained intron event had tissue specificity according to the isoCirc data. In several tissues (brain, testis, HEK293), the intron was primarily retained in circRNA transcripts. By contrast, in other tissues (adrenal, blood, smooth muscle), the intron was primarily spliced out in the circRNA transcripts (Fig. 4c). Additional examples of internal alternative splicing events corresponding to retained introns or skipped exons are illustrated in Supplementary Figs. 21 and 22.

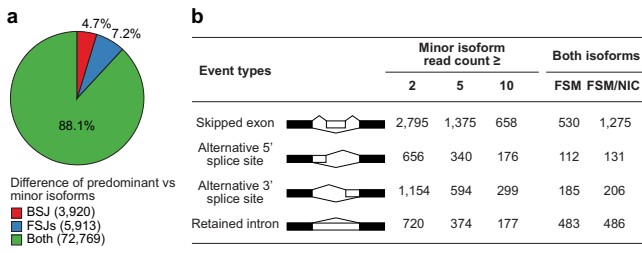

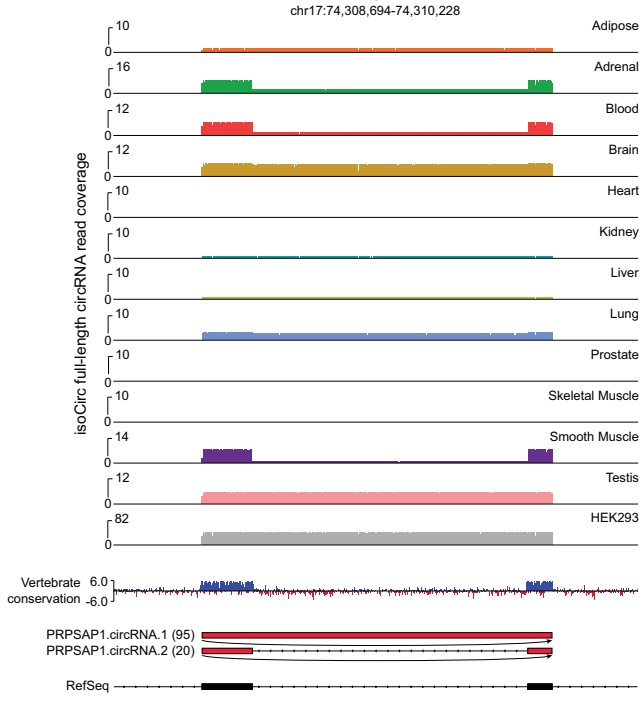

**Fig. 4 isoCirc discovery of alternative splicing events within circRNAs. a** Pie chart showing percentages of isoform pairs in which the predominant isoform (with highest median read count across 12 human tissues and HEK293 cell line for a given gene) had alternative splicing differences in BSJ only, FSJs only, or both compared to each of the other isoforms in the gene. Number of isoform pairs for each category is given in parentheses next to category name in the legend. **b** Summary table showing number of internal alternative splicing events within circRNAs corresponding to four major types of alternative splicing patterns, when requiring that the minor isoform had at least 2, 5, or 10 isoCirc reads. Number of internal alternative splicing events in which the splicing events of both isoforms had FSJs annotated as FSM only or FSM/NIC are represented in two rightmost columns. **c** isoCirc read coverage tracks for 12 human tissues and aggregated HEK293 replicates displaying the two most abundant circRNA isoforms of PRPSAP1 – PRPSAP1.circRNA.1 and PRPSAP1.circRNA.2, which had an alternative splicing event corresponding to a retained or spliced intron, respectively. A separate track displaying base-level conservation scores across vertebrates (phyloP 46-way) is supplied. Transcript structures and BSJs of PRPSAP1.circRNA.1 and PRPSAP1.circRNA.2 are shown using red boxes and black arrows. Total number of reads across all 12 human tissues and HEK293 replicates for each isoform is indicated next to the isoform identifier.

## Discussion

In recent years, long-read RNA-seq has gained substantial interest as an emerging technology for transcriptome analysis[30–32]. Although long-read RNA-seq has a higher error rate and lower throughput compared to short-read RNA-seq, the ability to sequence full-length transcripts is a key benefit and enables new applications not accessible by short-read RNA-seq. Strategies exist to improve the accuracy of long-read RNA-seq, by

generating raw reads that correspond to multiple copies of a template transcript sequence. In fact, circular consensus sequencing is a well-established approach to improve the accuracy of PacBio long reads[36,37]. Recently, a similar approach was developed for long-read nanopore RNA-seq of linear transcripts with the R2C2 method, in which linear transcripts are circularized followed by RCA and nanopore sequencing[38].

In this manuscript, we report isoCirc, a long-read sequencing strategy and companion software to determine full-length circRNA isoforms. The isoCirc experimental workflow combines RCA of template circRNA sequences and nanopore sequencing of RCA products. After calling consensus sequences from isoCirc raw reads, we use stringent criteria to remove potential false-positive detection of circRNA BSJs and full-length circRNA isoforms, which arises from consensus sequences with low mapping quality or spurious alignments. For BSJs involving novel splice sites to be reported, the filtering criteria are much stricter (as compared to BSJs involving known splice sites), requiring near-perfect alignment between the isoCirc consensus sequence and the corresponding genomic sequence around the BSJ. By performing isoCirc of 12 human tissues and one human cell line (HEK293), we generated a comprehensive catalog of full-length circRNA isoforms, including 107,147 isoforms supported by multiple isoCirc reads in at least one tissue or cell type. This dataset provides a useful resource that will facilitate studies into the biogenesis, regulation, and functions of circRNAs. The isoCirc software is available at https://github.com/Xinglab/isoCirc, and the catalog of full-length circRNA isoforms is available at https://genome.ucsc.edu/s/xinglab_chop/isoCirc.

As short-read RNA-seq is currently the standard approach for circRNA analysis, we compared isoCirc with short-read RNA-seq of circRNAs in several aspects. We showed that isoCirc substantially enriched circRNAs, with an over 70-fold enrichment of circRNA BSJ reads in isoCirc libraries compared to RNase R-treated short-read libraries (Supplementary Fig. 7). Short-read RNA-seq and isoCirc analyses of circRNA BSJs had a reasonable quantitative agreement, both for BSJ quantitation in individual samples (Supplementary Fig. 9) and for the numbers of BSJs identified across diverse human tissues (Fig. 3b). Furthermore, isoCirc had a comparable to slightly higher reproducibility among technical replicates, as compared to RNase R-treated short-read libraries (Supplementary Figs. 3 and 8).

We emphasize that a fundamental advantage of isoCirc (and long-read RNA-seq in general) is its ability to experimentally characterize full-length transcripts without computational inference. This feature is particularly useful for analyzing circRNAs, because the internal part (exons and FSJs) of full-length circRNA isoforms is shared with linear transcripts. This shared nature complicates computational inference, especially for regions within circRNAs far from the BSJs. Indeed, although computational methods to reconstruct full-length circRNAs from short-read RNA-seq data exist[25–29], these methods are unable to reconstruct long full-length circRNAs[25]. In a recent assessment of state-of-the-art reconstruction strategies, Zheng and colleagues concluded that short-read-based reconstruction methods can only be applied to short circRNAs (200–500 nt, depending on RNA-seq read length)[25]. In the present study, of the isoCirc isoforms classified as FSM or NIC for both the BSJ and FSJs, 41.7% and 8.7% were ≥500 and 1000 nt, respectively. In other words, almost half of these full-length circRNA isoforms identified by isoCirc in the 12 tissues were beyond the reach of short-read-based reconstruction methods, even under the best-case scenario for read length. This advantage of long-read sequencing is exemplified by the isoCirc discovery of many EIciRNAs, which are challenging to identify by short-read RNA-seq due to the large intron size of human genes. Collectively, these results highlight isoCirc as a useful long-read-based strategy for circRNA analysis.

In a bioRxiv preprint, Rahimi and colleagues described a strategy for nanopore sequencing of circRNAs[39]. Although both their method and isoCirc used nanopore long-read RNA-seq to analyze circRNAs, there are some important differences between these two methods and studies. First, unlike isoCirc, which amplifies and sequences full-length circRNAs, the method by Rahimi and colleagues does not guarantee that full-length circRNAs are sequenced, as circRNAs are nicked and linearized by gentle hydrolysis before nanopore sequencing. Indeed, their sequencing data showed that most (>87%) of their "circRNA mapping reads" were not full-length because they did not cross circRNA BSJs[39]. Second, the Rahimi et al. dataset had a high error rate of 6.8% and 6.4% per base for their human and mouse data, respectively, a typical error rate for standard nanopore 1D sequencing. By contrast, in isoCirc, the consensus sequences called from RCA products and used in the downstream analyses had a considerably lower error rate (Supplementary Fig. 1), with an average copy number of consensus sequence per raw read of 14.5 in the HEK293 data. Third, these two studies generated distinct datasets. Rahimi et al. analyzed one human brain sample and one mouse brain sample, from two male human donors and one male mouse, respectively. In the isoCirc study, we analyzed 12 human tissues and one cell line (HEK293). Each human tissue RNA was a pooled sample from tissues of multiple donors. Despite these differences, both studies present important methodological advances and highlight the value of nanopore long-read sequencing for circRNA analysis.

This work has some limitations. At the current sequencing depth, we expect that isoCirc is far from saturation, and more circRNA isoforms will be identified with deeper sequencing. This is a common issue for long-read RNA-seq, as the relatively low throughput and high per-sequence cost make it prohibitively expensive to deeply sample the transcriptome[30–32]. Additionally, in this work isoCirc was applied to whole cells and tissues. As the functions of circRNAs are coupled to their subcellular localization[3,5], in future studies, it would be useful to apply isoCirc to RNAs isolated from specific subcellular fractions. For example, isoCirc of chromatin-associated RNAs may identify new circRNA isoforms, such as EIciRNAs that are localized on chromatin and regulate gene transcription.

## Methods

**Cell culture conditions**. Human embryonic kidney cells (HEK293, ATCC #CRL-1573) were maintained at ~$3 \times 10^6$ cells per plate at 37 °C in a humidified 10% $CO_2$ atmosphere in Modified Eagle's Medium (Gibco) and 10% fetal bovine serum (Invitrogen), which had been filtered through a 0.2-um PES membrane filter (Nalgene Rapid-flow). HEK293 cells were used at passage 27 to 29. Cells were collected at approximately 80 to 90% confluency. Genomic DNA of HEK293 cells was authenticated by STR profiling. Cells were confirmed to be mycoplasma-free by the Lonza MycoAlert assay.

**Nucleic acid precipitation**. Acid-phenol: chloroform: isoamyl alcohol (125:24:1, pH 4.5) or phenol: chloroform: isoamyl alcohol (25:24:1, pH 8.0) was added to an equal volume of the solution intended for RNA or DNA precipitation, respectively. The mixture was vigorously shaken, incubated at room temperature for 5 min, and centrifuged (12,000 $g$, 15 min, 4 °C). Supernatant was transferred to a new tube and mixed with 0.1 volumes of sodium acetate (3 M, pH 5.2) and 0.01 μl of 20 mg/ml glycogen. The solution was combined with 2.5 volumes of ethanol or an equal volume of isopropanol, and the tube was inverted several times to mix. The sample was kept at −20 °C for at least 1 h until precipitation occurred. After centrifugation (12,000 $g$, 30 min, 4 °C), the pellet was washed once with 70% ethanol. After drying in air, nuclease-free $H_2O$ or THE RNA Storage Solution (Thermo Fisher Scientific, #AM7001) was added to dissolve DNA or RNA into solution.

**RNA extraction and preparation**. DNA LoBind tubes (Eppendorf) were used for all steps of RNA preparation and long-read library construction. To obtain RNA for long-read sequencing, HEK293 cells were lysed by using TRIzol Reagent (Invitrogen, #15596026, #15596018). Briefly, total RNA (20 μg) was extracted from HEK293 cells by adding 0.3–0.4 ml of TRIzol per $1 \times 10^5$–$10^7$ cells directly into the cell culture dish, in accordance with the manufacturer's instructions. Extracted

RNA was used immediately after extraction, or was stored at −80 °C in THE RNA Storage Solution and used within 3 days of extraction. Total RNA was treated with DNase I (Invitrogen, #AM2238) to remove DNA, in accordance with the manufacturer's instructions. After RNA extraction and DNase I treatment, RNA quality was assessed on an Agilent 2100 bioanalyzer (Agilent Technologies). Total RNA samples with RNA integrity number >9.8 were retained for circRNA enrichment. RNA and double-stranded DNA (dsDNA) concentrations were determined on a Qubit 2.0 fluorometer by using Qubit HS RNA and dsDNA kits (Life Technologies).

For long-read sequencing of human tissues, we obtained RNA samples (50 μg) of 12 human tissues (lung, adrenal, liver, kidney, adipose, skeletal muscle, smooth muscle, prostate, heart, brain, testis, blood [peripheral leukocytes]) from Clontech (Supplementary Table 4). The total RNA sample of each human tissue was a pooled sample extracted from tissues of multiple donors. Quality control information of extracted total RNA from human tissues was provided by the manufacturer. Subsequent library preparation procedures were the same for total RNAs from HEK293 cells and human tissues.

**CircRNA enrichment for long-read sequencing**. Total RNA (20–30 μg, concentration of ~1–2 μg/μl) was subjected to rRNA removal by using the RiboMinus Eukaryote Kit for RNA-seq (Thermo Fisher Scientific, #A1083708). Next, rRNA-depleted RNA samples (~1–2 μg) were subjected to digestion with RNase R (Epicentre, #RNR07250) at 37 °C for 30 min. Each 20-μl reaction contained 20 U of RNase R enzyme. Digested RNA was purified through AMPure beads (1.8× reaction volume) and eluted in 12 μl of RNase-free 10 mM Tris (pH 8.0). RNA was reverse transcribed using ProtoScript II Reverse Transcriptase (New England Biolabs [NEB], #EP0751), according to the manufacturer's protocol. A mixture of 10 μl of eluted RNA, 1 μl (100 pmol) of random hexamer, and 1 μl of dNTP mix (10 mM each) was incubated at 65 °C for 5 min and chilled on ice for 2 min. Enzyme mix containing 1 μl of RNase inhibitor (Invitrogen, #AM2694), 1 μl of ProtoScript II Reverse Transcriptase, 2 μl of 0.1 M DTT, and 4 μl of 5× RT Buffer was added to 12 μl of RNA template mix.

Reverse transcription was performed in the thermocycler by using the following protocol: 25 °C incubation for 5 min, 42 °C for 1 min, and 65 °C for 10 min. The cDNA was purified by using AMPure beads (1.8×) and eluted with 45 μl of Tris buffer (10 mM, pH 8.0). Mung Bean nuclease (NEB, #M0250L) with 3′ to 5′ nuclease activity was used to digest single strands of cDNA and RNA. Each 50-μl reaction included 44 μl of eluted cDNA, 5 μl of 10× buffer, and 1 μl of enzyme incubated at 30 °C for 30 min. Digested products containing circRNA/cDNA double-stranded hybrids were purified through (1.8×) AMPure beads and were eluted with 54 μl of Tris buffer (10 mM, pH 8.0). A 60-μl ligation mixture containing 52 μl of purified digestion product, 2 μl of SplintR ligase (NEB, #M0375), and 6 μl of 10× buffer was incubated at 25 °C for 20 min for the ligation reaction to occur.

**RCA reaction**. A 60-μl aliquot of ligation product from the previous step was diluted by an equal amount (60 μl) of nuclease-free H₂O, and then combined with 120 μl of phenol: chloroform: isoamyl alcohol at 25:24:1 (pH 8.0). For precipitation, 2.5 volumes of ethanol were added. The sample was kept at −20 °C for at least 1 h. Precipitated circDNA (~300 ng) was used as the template in the 200-μl RCA reaction, according to the protocol of the TempliPhi 100 Amplification Kit (GE Healthcare, #25-6400-10). The mixture was incubated at 30 °C for 18 h, followed by 65 °C for 10 min. Finally, DNA was precipitated as described in the 'Nucleic acid precipitation' section.

**Library generation for nanopore long-read sequencing**. The debranching reaction was performed as follows. First, 2.5 μg of RCA products were annealed in a thermocycler, in accordance with instructions provided with T7 Endonuclease I (NEB, #M0302). Next, 2.5 μl of T7 Endonuclease I was added to the reaction, mixed, and incubated at 37 °C for 1 h. Debranched products were purified through isopropanol precipitation (see 'Nucleic acid precipitation') and subjected to size selection, with fragments at 3–50 kb being selected through BluePippin (Sage Science, #BLF7510 gel).

After size selection, 1.5 μg of DNA was used to prepare libraries for long-read sequencing by the ONT sequencer, using the Ligation Sequencing Kit (Oxford Nanopore, #SQK-LSK109). For HEK293 cells, we prepared a total of six libraries from two biological replicates, each with three technical replicates (Supplementary Table 1). Biological replicates were constructed from RNA samples derived from different cell cultures and different RNA extractions. Technical replicates were constructed separately from RNA samples derived from the same cell culture and RNA extraction. For HEK293 cells, ~300 ng of each DNA library was used, and ~20 G bp of data were generated per flow cell (Oxford Nanopore, #FLO-MIN106D). For human tissues, ~20–25 μg of total RNA per tissue type was used, with sequencing on multiple flow cells (Oxford Nanopore, #FLO-MIN106D) to obtain a yield of ~30 G bp of data per tissue type (Supplementary Table 4).

**Library generation for Illumina short-read sequencing**. Short-read sequencing was performed by using total RNA extracted from HEK293 cells, using the same cell culture conditions and RNA extraction procedures as described above. RNA concentration and quality were determined with NanoDrop 2000 (Thermo Fisher Scientific), Qubit3.0, and Agilent TapeStation 4200 instruments. Two types of libraries were constructed: RNase R-treated libraries (3 biological replicates; R1, R2, R3) and poly(A)-selected libraries (3 biological replicates; S1, S2, S3) (Supplementary Table 2). For RNase R-treated libraries, 2 μg total RNA per replicate was incubated with 20 U RNase R enzyme at 37 °C for 30 min. In total, ~200 ng of RNase R-treated RNA was used to generate cDNA libraries following the TruSeq protocol (Illumina, San Diego, CA). For poly(A)-selected libraries, 2 μg of total RNA per replicate was used to prepare libraries by following the TruSeq protocol (Illumina). All six libraries were sequenced on an Illumina HiSeq 2500 sequencer at the University of California Los Angeles BSCRC High-Throughput Sequencing Core. Each library was subjected to 101-bp paired-end sequencing, with a sequencing depth of >50 M reads.

**Base-calling of raw nanopore long-read data**. Raw nanopore long-read data (fast5 format) were base-called using the Guppy base-calling software (version 2.1.3) (https://community.nanoporetech.com/downloads) with the following settings: *guppy_basecaller --input_path /path/to/raw_data --save_path /path/to/output_folder --flowcell FLO-MIN106 --kit SQK-LSK109*.

**Overview of isoCirc computational pipeline to identify high-confidence BSJs and full-length circRNA isoforms**. Details of the isoCirc computational pipeline are described in Supplementary Methods. Briefly, consensus sequences were called from each long read and mapped to the reference genome. Multiple consensus sequences may be called for each long read, and each consensus sequence may be aligned to multiple genomic locations. For each consensus sequence, isoCirc filtered out alignment records with low mapping quality, and then selected the optimal alignment record. Next, for each long read, isoCirc used multiple measures of individual consensus sequences to select the optimal consensus sequence. Then, isoCirc tried to identify a high-confidence BSJ and high-confidence FSJs from the selected optimal consensus sequence based on its optimal alignment record. Only consensus sequences with a high-confidence BSJ were considered as candidate circRNAs. Candidate circRNAs in which the BSJ and all identified FSJs were high-confidence were defined as 'full-length circRNA isoforms'.

**Annotation databases**. The human GRCh37/hg19 reference genome was used to perform read mapping and downstream analyses. Annotations for linear transcripts were extracted from the Ensembl GRCh37.87 gene annotation file (ftp://ftp.ensembl.org/pub/grch37/release-87/gtf/homo_sapiens/Homo_sapiens.GRCh37.87.chr_patch_hapl_scaff.gtf.gz). Annotations for repetitive elements from Repeat-Masker were retrieved from the UCSC Table Browser (https://genome.ucsc.edu/cgi-bin/hgTables) on 05/04/2019.

**Classification of known and novel circRNA BSJs identified by isoCirc**. We collected the chromosome ID and the start and end coordinates of each high-confidence BSJ identified by isoCirc. Next, we compared these BSJs with those of circRNAs from two existing annotation databases: circBase[33] (http://www.circbase.org) and MiOncoCirc[22] (https://mioncocirc.github.io). BSJs listed in either annotation database were classified as known. Otherwise, the BSJs were classified as novel.

**Categories of circRNA BSJs and FSJs**. Following the nomenclature of Tardaguila and colleagues[35], which defines the structures of linear transcripts identified by long-read sequencing, isoCirc used three categories to classify all identified circRNA BSJs and FSJs: Full Splice Match (FSM), Novel In Catalog (NIC), and Novel Not in Catalog (NNC). A given BSJ was considered to be FSM if it existed in any of the provided circRNA annotation databases. Otherwise, the BSJ was considered to be NIC if both the donor and acceptor back-splice sites were cataloged in the gene annotation file, or NNC if either the donor or acceptor back-splice site was novel.

Likewise, the full set of FSJs of a given circRNA was considered to be FSM if the combination of FSJs was consistent with any linear transcript in the gene annotation file. Otherwise, it was considered to be NIC if all forward-splice sites were cataloged in the gene annotation file, or NNC if any forward-splice site was novel. All single-exon circRNAs (with no FSJ) were considered to be FSM for the FSJ classification.

**Gene assignment for circRNA isoforms**. For each identified circRNA isoform, isoCirc collected all splice sites (back- and forward-splice sites). The strandness of a circRNA can be defined unambiguously by its BSJ, using the consensus dinucleotide motif of the back-splice sites. Among all genes having consistent strandness (plus or minus) with the BSJ, isoCirc selected the gene with the largest number of splice sites in common with the circRNA isoform as the assigned gene. In the rare case that multiple genes met this criterion, isoCirc assigned the circRNA isoform to multiple genes. If a circRNA isoform had no assigned gene based on the above procedure, then it was assigned to any gene(s) having consistent strandness and at least 1 bp overlap with the circRNA isoform in their genomic coordinates, where overlap was calculated by using the start and end coordinates of the gene and the circRNA isoform. The assigned gene of the circRNA isoform was set as

'NA' if no gene could be assigned after performing the steps above. For circRNA isoforms assigned to a given gene, we assigned a numeric identifier to each circRNA isoform, with 'circRNA.1' designated as the isoform with the highest median read count (or highest mean read count, if there was a tie for median) across the 12 tissues and HEK293 cell line.

**Comparison of circRNA read counts between short-read and isoCirc long-read datasets of HEK293 cells.** Paired-end Illumina sequencing data with 101-bp read length were generated from RNase R-treated libraries (R1, R2, R3) and poly(A)-selected libraries (S1, S2, S3). Reads from the six samples were pooled and aligned to the GRCh37/hg19 reference genome using BWA-MEM[40] with '-T 19' and default settings. To increase sensitivity, BSJs were next detected from alignment records of all reads, by using CIRI2[41] with Ensembl GRCh37.87 annotation and default settings. BSJ reads in each sample were counted by using the BSJ read list provided by CIRI2. Proportions of reads with BSJs (BSJ read counts divided by total read counts) of the three poly(A)-selected libraries and three RNase R-treated libraries (Supplementary Table 2) were compared to those of the six isoCirc libraries of HEK293 cells (Supplementary Table 1).

To quantitatively compare circRNA read counts between RNase R-treated short-read libraries and isoCirc long-read libraries, for any detected BSJ, we summed its read count from the short-read or long-read libraries, respectively. We then used BSJs with at least 2 reads in both types of libraries (short-read and isoCirc long-read) to assess concordance and calculate Spearman's rank correlation coefficient of read counts between these two types of libraries.

**Comparison of numbers of circRNA BSJs detected from eight human tissues between published short-read data and isoCirc long-read data.** We compared the numbers of circRNA BSJs detected from eight human tissues (brain, testis, kidney, prostate, liver, heart, lung, and skeletal muscle) shared between the isoCirc dataset and a published short-read circRNA dataset of human tissues[17] (accession number: BIGD ID: PRJCA000751). Specifically, Ji and colleagues constructed short-read RNA-seq libraries using total RNAs from human tissues treated with RiboMinus and RNase R. CircRNA BSJs were identified by using four different tools: CIRI2[41], DCC[42], MapSplice[43], and CircExplorer2[27]. CircRNA BSJs detected by at least two tools and supported by at least two independent short reads were considered bona fide BSJs. Across the eight human tissues shared between this published short-read dataset and the isoCirc long-read dataset, we compared the numbers of unique circRNA BSJs detected. For the short-read data, the number of circRNA BSJs detected from a given tissue was normalized by the total number of short reads in that tissue. Likewise, for the isoCirc long-read data, the number of circRNA BSJs detected from a given tissue was normalized by the total number of long reads in that tissue. Consistent with the analysis of short-read data, only circRNA BSJs supported by at least two independent isoCirc long reads were included in this analysis.

**Pairwise tissue comparison of circRNA isoforms.** To compare the abundance of full-length circRNA isoforms between human tissues, we performed a two-step statistical test using read counts of circRNA isoforms. For each pair of tissues, we collected all circRNA isoforms having at least 2 reads in at least one of the two tissues. Within each gene, we calculated the isoform proportion of each isoform (i.e., read count of the circRNA isoform divided by the total read count of all circRNA isoforms for that gene). For each gene with at least two circRNA isoforms, we performed a chi-square test using all isoforms' read counts, and adjusted $p$-values by the false-discovery rate (FDR) approach via the Benjamini–Hochberg procedure. We considered a gene to have differential proportions of circRNA isoforms between a pair of tissues if its FDR was ≤0.05, and at least one of its isoforms had a between-tissue difference of isoform proportion of ≥0.05.

For each gene found to have differential proportions of circRNA isoforms between a pair of tissues, we next sought to identify the specific isoforms that had significant differences in isoform proportions between tissues. For each isoform in the gene, we constructed a 2 × 2 table, consisting of the read counts of the isoform and the total read counts of all other isoforms in the gene. Then, we performed a Fisher's exact test using the 2 × 2 table for each isoform. An isoform was considered to have significant differences in isoform proportions between tissues if the Fisher's exact test $p$-value was ≤0.05 and the difference in isoform proportion between the two tissues was ≥0.05.

**Identification of tissue-stable and tissue-specific circRNA isoforms from multi-tissue isoCirc datasets.** We sought to identify circRNA isoforms with tissue-stable or tissue-specific isoform proportions across the 12-tissue isoCirc datasets. Only isoforms with at least two reads in at least one tissue were included in this analysis. For each gene with at least two circRNA isoforms, we calculated the proportion of each isoform across the 12 tissues based on isoCirc read counts. An isoform was classified as tissue-stable if the isoform proportion in every tissue was >0.5 and the isoform had at least two reads in every tissue.

To identify isoforms that were tissue-specific, we considered two cases: (i) genes in which all isoforms were exclusively detected in a single tissue, and (ii) genes in which isoform proportions were not homogenous across the 12 tissues (chi-square test, FDR ≤ 5%). By default, isoforms in case (i) were classified as tissue-specific.

For case (ii), within each gene, we selected isoform-tissue pairs in which the isoform proportion in the given tissue was significantly greater than the overall isoform proportion across the 12 tissues (i.e., sum of read counts of the isoform across all 12 tissues divided by sum of read counts of all isoforms across all 12 tissues) (one-tailed binomial test, FDR ≤ 5%). Among the selected pairs, we identified tissue-specific isoforms if an isoform-tissue pair met the following criteria: the difference in isoform proportion between the given tissue and all other tissues was ≥0.05, and the isoform had at least two reads in the given tissue.

Parental gene expression patterns across the 12 human tissues were determined using relevant short-read RNA-seq data from GTEx V8[44]. Specifically, median gene-level TPMs of the 12 tissues from GTEx were used to calculate Yanai's tissue-specificity index, $\tau$[45]. Genes with $\tau = 0$ were defined as housekeeping, and genes with $\tau = 1$ were defined as tissue-specific. To check whether tissue-specific circRNA isoforms were enriched in tissue-specific genes, we constructed a 2 × 2 table containing the numbers of tissue-specific or non-tissue-specific isoforms from tissue-specific or non-tissue-specific genes. Isoforms across the 12 tissues that had at least two reads in at least one tissue and were assigned to genes with at least two detected isoforms were represented in the table. We performed Fisher's exact test on the table, under the null hypothesis that the tissue-specific genes are equally likely to express tissue-specific isoforms as non-tissue-specific genes.

**Reporting summary.** Further information on research design is available in the Nature Research Reporting Summary linked to this article.

## Data availability
Raw data (fastq files) and processed data (abundance measurements) for Illumina short-read and nanopore long-read sequencing of circRNAs in total RNA extracted from HEK293 cells (6 replicates), and nanopore long-read sequencing of circRNAs in total RNA extracted from 12 human tissues, were uploaded to GEO under accession number GSE141693. The catalog of full-length circRNA isoforms can be accessed at https://genome.ucsc.edu:/s/xinglab_chop/isoCirc. Short-read circRNA datasets of human tissues were downloaded under accession number PRJCA000751 from BIGD. Median gene-level TPMs by tissue for GTEx V8 short-read RNA-seq data were downloaded from the GTEx portal (https://gtexportal.org/home/datasets).

## Code availability
The isoCirc software is available at https://github.com/Xinglab/isoCirc (v1.0.0, https://doi.org/10.5281/zenodo.4264644)[46].

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

## Acknowledgements

This work was supported by faculty startup funds from the Children's Hospital of Philadelphia (CHOP). The authors thank Zhicheng Pan for technical assistance.

## Author contributions

Y.X. and L.L. conceived the study; R.J.X., Yan G., L.L., and Y.X. designed the research and developed the methodology; Yuan G., R.W., B.L., and Y.W. contributed to analytic tools; R.J.X., Yan G., Yuan G., R.W., and Y.X. analyzed data; and K.E.K.-E. and Y.X. wrote the paper with input from all other authors.

## Competing interests

Y.X. is a scientific cofounder of Panorama Medicine.
