## [Peer Review File · Nature Communications]

Reviewers' comments:

Reviewer #1 (Remarks to the Author):

Circular RNAs are one category of non-coding RNAs with a relatively stable circular structure that could function as a biomarker. The existing circular RNA detection methods based on next-generation sequencing technology still suffer from technical issues such as the difficulty of assembly full-length circRNA sequences and their isoforms. Xin et al. here present a novel library preparation method following the third generation nanopore sequencing technology to produce longer reads with more enriched-on back-splicing junctions. They also designed a novel and accurate computational pipeline to detect circRNA based on this library. Their method shows a considerable level of reproducibility. Furthermore, they have tested this method on cell lines and tissues and detect numerous circRNA isoforms, for instance, circular isoforms of KDM1A. Overall, this method is innovative and could have strong applications for the circRNA research community. The manuscript is well-written and well-organized, that only several places need to be clarified or discussed in more detail.

Major comments:

1. By using the nanopore technology, could the author specify what is average read length for long reads generated from each sample? And what is the average length of consensus sequences after consensus calling? Does the author observe the variations of consensus sequences' length among biological/technical replicates that could lead to differences in circRNA detection?
2. The third-generation sequencing technology such as Nanopore could produce reads with a higher error rate than NGS sequencing reads. I was wondering could the authors evaluate the error rate level in long reads? If so, what is the possible impact of the sequence error rate on defining BSJ reads?
3. The authors found that over half of isoCirc identified circRNAs were not existing in known circRNA databases. Are these circRNAs expressed in relatively lower levels compared to the known ones?

Minor comments:

1. The Github code repository and GEO raw data are not available for review, please release, at least release to the reviewer.
2. For the Figure 2a and 2b, it is difficult to interpret the data. Could we speculate an 25% ~ 50% of BSJ reads shared between technical replicates? Could the authors present a more specific number of these similarities? It may also be better for the authors to label the biological replicates and technical replicates clearly.
3. In Supplementary Figure 5, a considerable part of BSJ showed much higher counts in long-read samples. Is this due to the higher sensitivity of isoCirc for circRNA identification? How about the correlations using normalized read counts?
4. In supplementary Figure 6, it is confusing to see that two columns bars with different colors are annotated as "isoCirc". Perhaps it could be annotated as "isoCirc biorep1" and "isoCirc biorep2"?
5. The authors identified many tissue-shift and stable circRNA isoforms. Are the shift circRNA isoforms enriched in tissue-specific genes and the stable circRNA isoforms enriched in house-keeping genes?

Reviewer #2 (Remarks to the Author):

Ruijiao Xin et alia describes a method of detecting circular ribonucleic acids (circRNAs) based on random-primed reverse transcription, ligation, rolling circle amplification, Oxford Nanopore sequencing and mapping of the resulting concatemers followed by bioinformatic assignment of them to a reference gene. The method can be useful in (more conveniently, possibly more precisely) determining the exonic composition of circRNAs. Given circRNA research popularity, the work may be cited well.

The manuscript is generally clearly written and is easy to understand. My major concerns about the

text are for the sparse referencing with several obvious citations not included and (critically) an extremely minimalistic background and particularly discussion. These sections are insufficiently unfolded to place the work in the overall picture and to understand its value. Discussions in the context of similar methods are missing, and the biological part also could include more insightful overview of the data.

My overall major concern is the novelty, both rolling circle long read generation (including for achieving high read coverage of the same sequence, e.g. PMID 30201725) and nanopore sequencing of circRNAs (e.g. [biorxiv.org/content/10.1101/567164v1](https://doi.org/10.1101/567164v1)) are not new. This does not disqualify the manuscript but necessitates careful referencing and comparison to these and similar methods, to demonstrate the advantages and limitations of the isoCirc technique. Further, results overviewing the new level of depth potentially offered by isoCirc are missing. What are the new features of the exonic circRNA composition that are revealed with isoCirc? How do these compare against short-read-based methods? What are the major tendencies in the circRNA isoform differences across the tissues?

I think the manuscript needs to include the comparisons to the other methods and the overviews of the new biological data that can be extracted with the method before it may be considered for publishing in Nature Communications.

Below I list points including the above considerations unfolded for the respective parts of the manuscript, as well as other comments and suggestions for improvement.

Abstract:

1. First sentence is not very clear. In my opinion, a stronger statement could be used. Such as that for circular RNAs, the exact exonic composition and sometimes order of the exons cannot be reliably or conveniently inferred from the short-read sequencing data.
2. It remains unclear from the abstract, how isoCirc works and what it achieves in general and compared to the state-of-the-art. Also, if the authors are mentioning circRNAs from particular tissues it would be good to outline what new quantitative and qualitative results were in these with isoCirc. This all appears as critical for a method paper.

Main:

3. Line 34, refs 2-6: could the authors please include a broader selection of excellent works and reviews on structure and function of circRNAs?
4. Line 35-36: this statement needs to be made somewhat milder. First, the full-length circRNA sequences could be and have been determined, to a certain precision, using short read sequencing. Second, 'hurdles' at the other levels are also difficult to resolve, the main arguably is circRNA sequence similarity to their linear counterparts, which is especially difficult to resolve in functional studies.
5. Line 39, 'internal alternative splicing structures': I suggest removing at least 'internal alternative'. I do not know if there are 'external' splicing structures as a defined term, and the fact that there are variants clearly points to the alternative splicing. Strictly, these are not splicing structures (which is in the RNA precursor), but post-splicing or exonic composition of mature (processed) RNA.
6. Line 41, refs 8-10: strictly, methods inferring this from the data exist, such as PMID 31435647 (probably needs to be cited here and definitely discussed somewhere in the manuscript).
7. Line 104 (and further), what was the average coverage in this case? Read count >2 refers to the physical reads and the sequence coverage was supposed to be higher? Please indicate it as it is important for the understanding of how precise the results are. ONT requires high coverage (up to 50) to provide high-confidence sequence, or mapping effects/bias can be high otherwise.
8. Line 176 see above in (6), lines 173-178 it would be interesting (and necessary for understanding of the capabilities of the method) to provide a comparison of these features (single-exon circRNA part, exon count/length etc) with the quantifications existing in the literature from the short read data.
9. Lines 179-201: I feel that the discussion and the context of the results are not very well developed. As a bare minimum, performance differences between short read isoform exonic composition inference (ideally published methods) and the results as by isoCirc are necessary to be included and discussed. How does isoCirc perform compared to the other long-read-based circRNA detection methods, such as those based on random cleavage, as in [biorxiv.org/content/10.1101/567164v1](https://doi.org/10.1101/567164v1) by Karim Rahimi et alia? It would be great to also have a more detailed, and more discussed in detail, analysis of the observed cross-tissue changes.

Online Methods:

10. Lines 206-211: cells of what confluency were collected? Critical for circRNA research.
 11. Lines 213-222: what was the reason behind using alkaline and not acidic phenol mixture for RNA?
 12. Lines 249-2656: why ProtoScript was used and not a reverse transcriptase without RNase H activity?
 13. Lines 356-358: 'If a circRNA isoform had no assigned gene based on the above procedure, then it was assigned to any gene(s) having consistent strandness and at least 1 bp overlap with the circRNA isoform,...': unclear what 1 bp overlap; 'strandness' is also not very well defined throughout. Did the authors mean that their assumption was that the circRNA precursor RNA originated from a transcript overlapping with the annotated features ('genes') in either one of the directions?
- Figures:
14. Figure 2 a,b: similar comparison to the short read data is missing. As mentioned earlier, 'read count' in this case is confusing. What was the actual read coverage?
 15. Figure 2c: what is the reason behind such a high variance between the replicates? Axis title is confusing, it most likely is 'Replicate', not 'Number of replicates'.
 16. Figure 2e: very confusing description of 'read count'. Please clearly make a distinction between physical reads and concatemer read coverage.
 17. Figure 3: it may be OK to include these visualisations, but I do not find the presented information particularly insightful. What exonic feature differences were observed across the tissues? How do these differences compare to the short-read inferred structure of circRNAs (solely based on back-spliced junctions and on back-spliced junctions plus isoform prevalence inference)? What are the main trends of change across the tissues? Are the changes in alternative splice sites, alternative exon inclusion, exon skipping or back-splicing at different points present?
 18. Main figures and Supplementary figures need to be re-worked to include consistent (and always readable) font size/type and have proportional elements. Currently, at an anticipated published size, nearly all important labels and many of the graphical elements will be illegible.

Reviewer #3 (Remarks to the Author):

Xin et al described isoCirc, an experimental protocol and a companion computational pipeline to sequence circular RNAs (circRNA) with Oxford nanopore (ONT). While standard short-read RNA-seq can identify circRNA, the technology requires substantial sequencing depth and can't obtain full-length structure. This manuscript addresses both issues. I am not an expert on circRNAs or experimental protocols. I will focus on the computation and analysis part of this manuscript.

1) I can access the UCSC custom track provided by the authors. However, their code repository seems private. I am unable to access it. The circRNAs the authors found have been deposited to GEO, but the data are also private. I would appreciate if the authors can upload the source code and the circRNAs in HEK293 as supplementary data (or put them on a department FTP server). This will help me to evaluate the results.

2) A key question I have is: how many circRNAs the authors found are real and how many are transcription noises? Fig S1/S2 suggest that two replicates are more consistent if we increase the count threshold to 3. Fig 2c shows the majority of circRNAs found in one replicate only are "novel". Are these really novel? Can authors find a way to estimate the fraction of real circRNAs?

Here are a few of my thoughts, though I am not sure if they work. a) Replot Fig. 2a, 2b, 2c, 2d, 3a, Fig S1 and S2 for read count threshold 1 through 4. This will give readers an idea about whether the results are sensitive to the count threshold. If the results are sensitive to the threshold, the authors will probably get more isoforms with deeper sequencing. Is that right? b) The authors are looking for canonical splice signals around back-splicing junctions (BSJs) within a window. Is it harder (e.g. more distant to alignment ends) to find splice signals around less confident and novel BSJs? c) Some (not

all) BSJs are formed due to inverted repeats in flanking introns. Have the authors checked this feature? How many BSJs have inverted repeats nearby? Is the fraction of inverted repeats reduced among novel BSJs? The general idea is to stratify a biologically meaningful feature by the confidence level of BSJs. The authors have done that in Fig S1 and S2 but I think they should do more to understand the quality of their data.

3) If a circRNA doesn't have a forward-splicing junction (FSJ), i.e. it consists of a single exon, will it be counted as full splice match (FSM)? Perhaps that is a new category for FSJ?

Minor comments:

4) Supplementary Methods is not long and does not contain figures. I think it can be moved to Online Methods.

5) In Fig. 1a/1b, Fig S1 and S2, please write the exact "Degree of similarity" numbers in the heatmap. There should be enough space to put these numbers in the plots. It is difficult for me to tell what the similarity is by looking at the colors.

Detailed Responses to Reviewer Comments

Reviewer #1:

Circular RNAs are one category of non-coding RNAs with a relatively stable circular structure that could function as a biomarker. The existing circular RNA detection methods based on next-generation sequencing technology still suffer from technical issues such as the difficulty of assembly full-length circRNA sequences and their isoforms. Xin et al. here present a novel library preparation method following the third generation nanopore sequencing technology to produce longer reads with more enriched-on back-splicing junctions. They also designed a novel and accurate computational pipeline to detect circRNA based on this library. Their method shows a considerable level of reproducibility. Furthermore, they have tested this method on cell lines and tissues and detect numerous circRNA isoforms, for instance, circular isoforms of KDM1A. Overall, this method is innovative and could have strong applications for the circRNA research community. The manuscript is well-written and well-organized, that only several places need to be clarified or discussed in more detail.

We thank the reviewer for the highly positive comments about our work. The reviewer considers our manuscript “well-written and well-organized”, and the isoCirc method “innovative” with “strong applications for the circRNA research community”. The reviewer brought up a few additional points for clarification or further discussion. We have addressed these requests in the revised manuscript, as detailed below.

Major comments:

1. By using the nanopore technology, could the author specify what is average read length for long reads generated from each sample? And what is the average length of consensus sequences after consensus calling? Does the author observe the variations of consensus sequences' length among biological/technical replicates that could lead to differences in circRNA detection?

To summarize this information, we now include a new supplementary figure in the revised manuscript (**Supplementary Fig. 2**). Focusing on isoCirc reads for which consensus sequences were called and high-confidence BSJs were detected, we found that the average length of raw isoCirc reads was ~4 k nt or slightly longer, whereas the average length of consensus sequences was ~300 nt or slightly longer (depending on tissue type). Overall, the distributions of average raw read length and consensus sequence length were comparable among the 12 tissues and 6 HEK293 replicates, although there was some modest variation among different tissues.

Supplementary Fig. 2. Comparison of raw isoCirc read lengths and consensus sequence lengths across six HEK293 libraries and 12 human tissues.

a Violin plots showing distributions of lengths (in bp) for raw isoCirc reads across six HEK293 libraries and 12 human tissues. isoCirc raw reads for which consensus sequences were called and high-confidence BSJs were detected are represented in the plots.

b Violin plots showing distributions of lengths (in bp) for called consensus sequences across six HEK293 libraries and 12 human tissues. Consensus sequences for which high-confidence BSJs were detected are represented in the plots.

2. The third-generation sequencing technology such as Nanopore could produce reads with a higher error rate than NGS sequencing reads. I was wondering could the authors evaluate the error rate level in long reads? If so, what is the possible impact of the sequence error rate on defining BSJ reads?

The higher error rate of nanopore RNA-seq reads certainly poses a challenge. We took great care in this work to minimize the error rate and its effects on circRNA analyses, using a combination of computational and experimental procedures. Computationally, we used stringent criteria to remove potential false-positive detection of circRNA BSJs and full-length circRNA isoforms due to isoCirc reads with low mapping quality or spurious alignments. Experimentally, we generated rolling circle amplification (RCA) products of circRNA isoforms. RCA products contain multiple copies of a template circRNA sequence. Consensus sequences called from RCA products are expected to have a reduced error rate, thus improving downstream analyses. To confirm this, for all isoCirc reads for which consensus sequences were called and mappable, we estimated the error rates of raw reads and of consensus

sequences called from raw reads with different copy numbers. Raw reads had the highest error rate. As expected, we observed a copy number-dependent decrease in the error rate of consensus sequences, with consensus sequences called from raw reads with >10 copies having the lowest error rate. We include this information in the new **Supplementary Fig. 1** of the revised manuscript:

Supplementary Fig. 1. Comparison of error rates for raw isoCirc reads and called consensus sequences across six HEK293 libraries.

Violin plots showing distributions of error rates for raw isoCirc reads and for consensus sequences called from 2, 3, 4, 5, 6 to 10, and more than 10 copies of a template circRNA sequence across six HEK293 libraries. isoCirc reads that had consensus sequences called and were mappable to the human genome are represented in the plots.

3. The authors found that over half of isoCirc identified circRNAs were not existing in known circRNA databases. Are these circRNAs expressed in relatively lower levels compared to the known ones?

We apologize for the potential confusion regarding Fig. 2c and its legend. The intention of Fig. 2c is to show that the set of circRNA BSJs detected in both HEK293 biological replicates contains a higher percentage of known BSJs documented in existing circRNA databases (85.1%), as compared to the set of BSJs detected in only one of the two HEK293 biological replicates (46.9%). This result is expected. We have revised our main text and the figure legend to clarify this point. To address the reviewer's question, we have also plotted the cumulative distribution of BSJ read counts for known or novel BSJs detected in both replicates or only one replicate. As expected, known BSJs have higher read counts than novel BSJs, although the difference is modest, especially for the comparison between known vs novel BSJs detected in only one replicate. Notably, novel BSJs detected in both replicates have higher read counts compared to known BSJs detected in only one replicate. These results are now summarized in a new supplementary figure (**Supplementary Fig. 11**):

Supplementary Fig. 11. Cumulative distribution plots of read counts in 2 replicates for high-confidence BSJs identified in HEK293 cells.

a Cumulative distribution plots of read counts in 2 replicates for BSJs at read count ≥ 2 . BSJs were classified based on their annotation status and level of detection across biological replicates of HEK293 cells, as follows: BSJ is known and detected in both replicates (red); BSJ is novel and detected in both replicates (blue); BSJ is known and detected in only one replicate (green); and BSJ is novel and detected in only one replicate (purple).

b Cumulative distribution plots of read counts in 2 replicates for BSJs at read count ≥ 3 . Details are as in **(a)**.

Minor comments:

1. The Github code repository and GEO raw data are not available for review, please release, at least release to the reviewer.

In the original submission, the GEO dataset was private. We shared a token for reviewer access, but apparently this mechanism did not allow the reviewers to access the companion raw isoCirc data deposited into SRA. There is no good mechanism to share access to a private Github repository. Although we had uploaded the entire software package as a Supplementary File to our manuscript, both reviewers 1 and 3 missed this file. To address these issues, we have now made both the Github repository and GEO/SRA data public, so the reviewers should have no problem accessing the software and data.

2. For the Figure 2a and 2b, it is difficult to interpret the data. Could we speculate an 25% ~ 50% of BSJ reads shared between technical replicates? Could the authors present a more specific number of these

similarities? It may also be better for the authors to label the biological replicates and technical replicates clearly.

We appreciate the reviewer's suggestions. We have added the actual value for the degree of similarity among a given pair of libraries (i.e., number of BSJs or full-length isoforms detected in both libraries, divided by total number of unique BSJs or full-length isoforms detected in either library) to Figs. 2a and 2b, as well as to Supplementary Figs. 3 and 4. As illustrated in these figures, the degree of similarity among replicates was higher when we required that identified BSJs (Supplementary Fig. 3) or full-length isoforms (Supplementary Fig. 4) be supported by a larger number of isoCirc reads. As expected, technical replicates from the same biological replicate had slightly higher similarity than technical replicates from different biological replicates. Additionally, to address the reviewer's suggestion about sample labels, we have replaced the library IDs in the original figure with more informative sample labels showing the biological replicate number plus the technical replicate number (e.g., rep1_1). We have made this update to all figures and supplementary figures involving these six HEK293 replicates.

3. In Supplementary Figure 5, a considerable part of BSJ showed much higher counts in long-read samples. Is this due to the higher sensitivity of isoCirc for circRNA identification? How about the correlations using normalized read counts?

As illustrated in Supplementary Fig. 7, the isoCirc experimental protocol substantially enriches circRNAs, with >70-fold enrichment of circRNA BSJ reads in isoCirc libraries vs RNase R-treated Illumina libraries. Therefore, it is not surprising that certain BSJs have few reads in the Illumina libraries but many more reads in the isoCirc libraries (especially as none of these libraries are expected to reach saturation at the present sequencing depth). Nonetheless, at read count ≥ 2 , we observed a Spearman correlation of 0.5 between the short-read data and isoCirc data (Supplementary Fig. 9 of the revised manuscript), indicating a reasonable agreement between these two distinct methods for BSJ quantitation. To calculate the correlation coefficient between two libraries, there is no need to normalize read counts by library depth, because such normalization will not alter the Spearman (or Pearson) correlation coefficient.

4. In supplementary Figure 6, it is confusing to see that two columns bars with different colors are annotated as "isoCirc". Perhaps it could be annotated as "isoCirc biorep1" and "isoCirc biorep2"?

In accordance with the reviewer's suggestion, we have revised this figure (now Supplementary Fig. 7 of the revised manuscript) to change the labels of the isoCirc samples.

5. The authors identified many tissue-shift and stable circRNA isoforms. Are the shift circRNA isoforms enriched in tissue-specific genes and the stable circRNA isoforms enriched in house-keeping genes?

To address this question, we partitioned 68,527 circRNA isoforms (from genes with ≥ 2 detected circRNA isoforms not assigned to multiple genes) from 12 human tissues based on two categories: **1**) circRNA expression pattern (tissue-stable, tissue-specific, other), and **2**) parental gene expression pattern (housekeeping, tissue-specific, other) (**Table R1** below). We determined parental gene expression patterns across the 12 human tissues using relevant short-read RNA-seq data from GTEx V8. The median gene-level TPMs of 12 tissues from GTEx were used to calculate Yanai's tissue-specificity index, *tau* (Bioinformatics 2005; 21(5):650-659). Genes with *tau* = 0 were defined as housekeeping, and genes with *tau* = 1 were defined as tissue-specific.

Table R1: Summary of parental gene expression patterns of 68,527 circRNA isoforms from 12 human tissues, categorized by circRNA expression pattern.

CircRNA expression pattern	Parental gene expression pattern			Total
	Housekeeping	Tissue-specific	Other	
Tissue-stable	1	0	98	99
Tissue-specific	26	874	2,461	3,361
Other	614	2,329	62,124	65,067
Total	641	3,203	64,683	68,527

Based on **Table R1**, we found that the circRNA expression pattern was not globally independent of the parental gene expression pattern ($p < 2.2e-16$, chi-square test). As a post-hoc test, we computed adjusted standardized residuals to investigate cells in this table that made statistically significant contributions to this observation. We found that the number of tissue-specific circRNA isoforms from tissue-specific genes was significantly higher than random expectation ($p < 2.2e-16$). We have described this result in the revised manuscript.

Reviewer #2:

Ruijiao Xin et alia describes a method of detecting circular ribonucleic acids (circRNAs) based on random-primed reverse transcription, ligation, rolling circle amplification, Oxford Nanopore sequencing and mapping of the resulting concatemers followed by bioinformatic assignment of them to a reference gene. The method can be useful in (more conveniently, possibly more precisely) determining the exonic composition of circRNAs. Given circRNA research popularity, the work may be cited well.

The manuscript is generally clearly written and is easy to understand. My major concerns about the text are for the sparse referencing with several obvious citations not included and (critically) an extremely minimalistic background and particularly discussion. These sections are insufficiently unfolded to place the work in the overall picture and to understand its value. Discussions in the context of similar methods are missing, and the biological part also could include more insightful overview of the data.

My overall major concern is the novelty, both rolling circle long read generation (including for achieving high read coverage of the same sequence, e.g. PMID 30201725) and nanopore sequencing of circRNAs (e.g. [biorxiv.org/content/10.1101/567164v1](https://doi.org/10.1101/567164v1)) are not new. This does not disqualify the manuscript but necessitates careful referencing and comparison to these and similar methods, to demonstrate the advantages and limitations of the isoCirc technique.

Further, results overviewing the new level of depth potentially offered by isoCirc are missing. What are the new features of the exonic circRNA composition that are revealed with isoCirc? How do these compare against short-read-based methods? What are the major tendencies in the circRNA isoform differences across the tissues? I think the manuscript needs to include the comparisons to the other methods and the overviews of the new biological data that can be extracted with the method before it may be considered for publishing in Nature Communications.

Below I list points including the above considerations unfolded for the respective parts of the manuscript, as well as other comments and suggestions for improvement.

We thank the reviewer for the positive comments about our work. The reviewer commented that “the method can be useful in determining the exonic composition of circRNAs. Given circRNA research popularity, the work may be cited well.” The reviewer also commented that “the manuscript is generally clearly written and is easy to understand.” The reviewer listed some major concerns of this manuscript. Here, we briefly summarize our responses to these major concerns.

One major concern of the reviewer was “for the sparse referencing with several obvious citations not included and (critically) an extremely minimalistic background and particularly discussion...Discussions in the context of similar methods are missing, and the biological part also could include more insightful overview of the data.” We agree with the reviewer on the brevity of our original manuscript, and would like to explain the reason for that. We initially submitted our manuscript as a Brief Communication to *Nature Methods*, which strictly limits the manuscript length (1000-1500 words), abstract length (3 sentences, 70 words), number of figures (2-3), and number of citations (20). After communications among editors of the two journals, we were advised that our manuscript would be transferred without reformatting to *Nature Communications* for review. Consequently, our reviewed manuscript was considerably abbreviated compared to typical manuscripts submitted to *Nature Communications*, with minimal background and discussion.

In our revision, we have greatly expanded the manuscript to place our work within the context of the rich circRNA field. We have performed additional analyses and included substantial new data, including a new section in the main text on the features of the exonic circRNA composition revealed by isoCirc. We

have included additional comparisons and discussions of existing methods (short-read- or long-read-based) for circRNA sequencing, including the bioRxiv preprint by Rahimi et al. (2019), highlighting the novelty and value of isoCirc. We are pleased and hope the reviewer will agree that these revisions have substantially improved the quality, clarity, and impact of our manuscript.

Another major concern expressed by the reviewer was the novelty of the work. The reviewer commented that “both rolling circle long read generation (including for achieving high read coverage of the same sequence, e.g. PMID 30201725) and nanopore sequencing of circRNAs (e.g. [biorxiv.org/content/10.1101/567164v1](https://doi.org/10.1101/567164v1)) are not new”. The reviewer is correct that both rolling circle amplification and nanopore sequencing of circRNAs have been described in either peer-reviewed or preprinted publications. However, the combination of these two techniques for sequencing full-length circRNAs is novel and has not been described in the literature. In this manuscript, we provide: **1)** a novel sequencing strategy combining rolling circle amplification and nanopore long-read sequencing for characterizing full-length circRNA isoforms, **2)** companion software (publicly available on Github) for rigorous analyses of isoCirc data, and **3)** a comprehensive catalog of full-length circRNA isoforms across diverse human tissues as a resource. These are important novel contributions to the circRNA field.

Collectively, we believe that this work has a high level of innovation and impact, and has the potential to be widely read and cited, as noted by the reviewer. We realize that part of the reviewer’s concern regarding novelty might come from the insufficient comparison to and discussion of existing methods, which we have rectified in this revision by substantially expanding the manuscript. For example, there are some key methodological differences between isoCirc and the nanopore sequencing method described by Rahimi et al. (2019), which we have discussed in detail in the revised manuscript (please see our response to the reviewer’s comment #9 below).

Our detailed point-by-point responses to each of the reviewer’s comments can be found below.

Abstract:

1. First sentence is not very clear. In my opinion, a stronger statement could be used. Such as that for circular RNAs, the exact exonic composition and sometimes order of the exons cannot be reliably or conveniently inferred from the short-read sequencing data.

We agree with the reviewer and have modified the abstract to make a stronger statement about this point. The new abstract can be seen in our response to the reviewer’s comment #2 below.

2. It remains unclear from the abstract, how isoCirc works and what it achieves in general and compared to the state-of-the-art. Also, if the authors are mentioning circRNAs from particular tissues it would be good to outline what new quantitative and qualitative results were in these with isoCirc. This all appears as critical for a method paper.

We appreciate the reviewer’s suggestion. As noted above, this manuscript was initially prepared and submitted as a Brief Communication, and the abstract had a strict limit of 3 sentences and 70 words. We have now substantially expanded the abstract to make it more informative. The new abstract is as follows:

“Circular RNAs (circRNAs) have emerged as an important class of functional RNA molecules. Short-read RNA sequencing (RNA-seq) is a widely used strategy to identify circRNAs. However, an inherent limitation of short-read RNA-seq is that it does not experimentally determine the full-length sequences and exact exonic compositions of circRNAs. Here, we report isoCirc, a novel strategy for sequencing full-length circRNA isoforms, using rolling circle amplification followed by nanopore long-read sequencing. We describe an integrated computational pipeline to reliably characterize full-length circRNA isoforms using isoCirc data. Using isoCirc, we generated a

comprehensive catalog of 107,147 full-length circRNA isoforms across 12 human tissues and one human cell line (HEK293), including 40,628 isoforms ≥ 500 nt in length. We identified widespread alternative splicing events within the internal part of circRNAs, including 720 retained intron events corresponding to a class of exon-intron circRNAs (EiCIRNAs). Collectively, isoCirc and the companion dataset provide a novel strategy and resource for studying circRNAs in human transcriptomes.”

Main:

3. Line 34, refs 2-6: could the authors please include a broader selection of excellent works and reviews on structure and function of circRNAs?

As this manuscript was initially submitted as a Brief Communication, we were constrained by a strict limit of 20 citations. We have now substantially expanded the introduction section to discuss a broad selection of research articles and reviews on the structure and function of circRNAs, as suggested by the reviewer.

4. Line 35-36: this statement needs to be made somewhat milder. First, the full-length circRNA sequences could be and have been determined, to a certain precision, using short read sequencing. Second, ‘hurdles’ at the other levels are also difficult to resolve, the main arguably is circRNA sequence similarity to their linear counterparts, which is especially difficult to resolve in functional studies.

We have revised and expanded the text to make the statement milder and more balanced. The text now reads:

“Despite the tremendous successes of short-read RNA-seq studies of circRNAs, an inherent limitation of this approach is that short-read RNA-seq does not experimentally determine the full-length sequences and internal alternative splicing events within circRNAs¹³. Very few circRNAs have been functionally characterized, and functional studies of circRNAs substantially benefit from knowledge of full-length circRNA sequences¹³. For example, to identify circRNAs that act as sponges for microRNAs or RNA-binding proteins, it is essential to know their full-length sequences (as opposed to the BSJs alone)⁹. Similarly, inferring the protein products translated from circRNAs requires full-length circRNA sequences¹¹. To fill this gap, several computational methods have been developed to reconstruct full-length circRNAs from short-read RNA-seq data^{25, 26, 27, 28, 29}. However, these methods are only applicable to short circRNAs (200-500 nt, depending on RNA-seq read length), and are unable to interrogate longer full-length circRNAs²⁵.”

5. Line 39, ‘internal alternative splicing structures’: I suggest removing at least ‘internal alternative’. I do not know if there are ‘external’ splicing structures as a defined term, and the fact that there are variants clearly points to the alternative splicing. Strictly, these are not splicing structures (which is in the RNA precursor), but post-splicing or exonic composition of mature (processed) RNA.

We adopted the phrase “internal alternative splicing structures” from one of the first papers that used short-read RNA-seq data to reconstruct full-length circRNAs, entitled “Comprehensive identification of internal structure and alternative splicing events in circular RNAs” (Nat Commun. 2016;7:12060). In this context, “internal” was used to distinguish these splicing events from back-splicing events, which can also be alternative. We should also note that in the reference suggested in the reviewer’s comment #6 below (PMID 31435647), the phrase “internal splicing patterns” was used in the abstract. Therefore, based on the prior literature and the reviewer’s suggestion, we have modified “internal alternative splicing structures” to “internal alternative splicing events” in the revised manuscript.

6. Line 41, refs 8-10: strictly, methods inferring this from the data exist, such as PMID 31435647 (probably needs to be cited here and definitely discussed somewhere in the manuscript).

We have expanded our citation list to include additional methods that reconstruct full-length circRNAs from short-read RNA-seq data (e.g., PMID 31435647 as suggested by the reviewer).

7. Line 104 (and further), what was the average coverage in this case? Read count >2 refers to the physical reads and the sequence coverage was supposed to be higher? Please indicate it as it is important for the understanding of how precise the results are. ONT requires high coverage (up to 50) to provide high-confidence sequence, or mapping effects/bias can be high otherwise.

We apologize for the confusion. The term “read count” in our paper refers to the number of independent nanopore reads supporting a given circRNA BSJ or full-length isoform. This is distinct from the copy number of a given template circRNA sequence within the nanopore read (i.e., RCA product). We have modified our text to clarify this point. As shown in the new **Supplementary Fig. 1**, the raw reads had the highest error rate. As expected, we observed a copy number-dependent decrease in the error rate of consensus sequences, with consensus sequences called from raw reads with >10 copies having the lowest error rate. All raw reads used in further analyses contained at least 2 copies of consensus sequences.

Supplementary Fig. 1. Comparison of error rates for raw isoCirc reads and called consensus sequences across six HEK293 libraries.

Violin plots showing distributions of error rates for raw isoCirc reads and for consensus sequences called from 2, 3, 4, 5, 6 to 10, and more than 10 copies of a template circRNA sequence across six HEK293 libraries. isoCirc reads that had consensus sequences called and were mappable to the human genome are represented in the plots.

As demonstrated in this figure, the use of RCA and consensus sequences called from RCA products improves downstream data analyses. Additionally, we used stringent criteria to remove potential false-positive detection of circRNA BSJs and full-length circRNA isoforms due to isoCirc reads with low

mapping quality and spurious alignments. This process helped to ensure high-confidence detection of circRNA BSJs and full-length circRNA isoforms. We should note that the predominant nanopore RNA-seq methods used today are 1D RNA-seq and direct RNA-seq. In both cases, the copy number per raw read is 1.

We are not entirely sure about the context of the 50x coverage required by ONT, as noted by the reviewer, but we guess that this refers to a requirement regarding ONT-based genome sequencing. In transcriptomics, we typically do not see this kind of metric used to describe sequencing coverage, because the transcriptome is not uniform. Here, we used read count ≥ 2 as the threshold for the number of independent nanopore reads supporting a given transcript. This is the conventional threshold used in the RNA-seq literature to select reliable events (e.g., see *Nucleic Acids Res.* 2010 38(14):4570-8). We also present data based on a more stringent threshold (read count ≥ 3).

8. Line 176 see above in (6), lines 173-178 it would be interesting (and necessary for understanding of the capabilities of the method) to provide a comparison of these features (single-exon circRNA part, exon count/length etc) with the quantifications existing in the literature from the short read data.

The reviewer asked for a more comprehensive comparison of features learned by isoCirc to those learned from short-read RNA-seq of circRNAs. A fundamental advantage of long-read RNA-seq is that it can characterize full-length isoforms experimentally, without the need for computational inference. This feature is particularly useful for analyzing circRNAs, because the internal part (exons and forward-splice junctions) of full-length circRNA isoforms are shared with linear transcripts. Therefore, short-read RNA-seq cannot be considered the gold-standard for comparison to long-read RNA-seq, especially in the discovery and quantitation of full-length isoforms. In fact, to our knowledge, in the published literature on long-read RNA-seq, a comparison to short-read data regarding isoform features has not been customarily performed as a standard for publication (e.g., *Genome Res.* 2018;28(3):396-411; *PNAS* 2018; 115(39):9726-9731; [biorxiv.org/content/10.1101/567164v1](https://doi.org/10.1101/567164); the latter two are the R2C2 paper and the Rahimi et al. (2019) bioRxiv preprint referenced by the reviewer, neither of which made a comparison to short-read based isoform results).

Nonetheless, we have compared our results with short-read based results in several aspects. In the original manuscript, we showed that short-read and isoCirc-based analyses of circRNA BSJs had a reasonable quantitative agreement, both for BSJ quantitation in individual samples (Supplementary Fig. 9 now) and for numbers of BSJs identified across diverse human tissues (Fig. 3b). We also showed that isoCirc substantially enriches circRNAs, with >70-fold enrichment of circRNA BSJ reads in isoCirc libraries vs RNase R-treated short-read libraries (Supplementary Fig. 7 now). In the revision, we have performed a new analysis to show that, among technical replicates, isoCirc has comparable-to-higher reproducibility as compared to RNase R-treated short-read libraries at the same read-count threshold (see our response to the reviewer's comment #14 below).

Although computational methods have been developed to reconstruct full-length circRNAs from short-read RNA-seq data, these methods are unable to reconstruct long full-length circRNAs. In a recent publication by Dr. Fangqing Zhao (developer of the widely used CIRI and CIRI-AS for short-read based circRNA analysis), the authors comprehensively assessed the state-of-the-art for reconstructing full-length circRNAs from short-read RNA-seq data (*Genome Med.* 2019;11:2). The conclusion was that short-read-based reconstruction methods are only applicable to short circRNAs (200-500 nt, depending on RNA-seq read length). In our work, of the isoCirc isoforms classified as FSM or NIC for both the BSJ and FSJs, 41.7% and 8.7% were ≥ 500 nt and 1,000 nt, respectively, with the longest isoform being 2,039 nt. In other words, almost half of these full-length circRNA isoforms identified by isoCirc are beyond the reach of short-read-based reconstruction methods, even under their best-case scenario for RNA-seq read

length. This is a fundamental distinction between isoCirc and short-read RNA-seq. We have revised and expanded the main text to emphasize this point.

Additionally, we have performed new analyses and included a new section in the main text and a new multi-panel figure (**Fig. 4**), to report the features of the exonic circRNA composition revealed by isoCirc and a comparison to previous short-read based results. This is described in detail in our response to the reviewer's comment #9 below.

9. Lines 179-201: I feel that the discussion and the context of the results are not very well developed. As a bare minimum, performance differences between short read isoform exonic composition inference (ideally published methods) and the results as by isoCirc are necessary to be included and discussed. How does isoCirc perform compared to the other long-read-based circRNA detection methods, such as those based on random cleavage, as in [biorxiv.org/content/10.1101/567164v1](https://doi.org/10.1101/567164v1) by Karim Rahimi et alia? It would be great to also have a more detailed, and more discussed in detail, analysis of the observed cross-tissue changes.

To address these comments, we have performed new analyses. The revised manuscript includes a new section in the main text and a new multi-panel figure (Fig. 4), to report the features of the exonic circRNA composition revealed by isoCirc and a comparison to previous short-read-based results. Briefly, for genes with multiple circRNA isoforms identified by isoCirc across the 12 human tissues and the HEK293 cell line, we compared the predominant (most abundant) isoform to each of the other isoforms in the gene. We identified isoform pairs with alternative splicing differences in BSJ only, FSJs only, or both. Focusing on the internal part of circRNAs, we identified >5,000 alternative splicing events corresponding to four major types of alternative splicing patterns (skipped exon, alternative 5' splice site, alternative 3' splice site, retained intron). We compared these results to those on HEK293 cells from a short-read-based analysis of internal alternative splicing events in circRNAs (Nat Commun. 2016; 7:12060). For most types of alternative splicing patterns, isoCirc reported several (2.4-4.3) times more events. A notable exception is for retained introns, for which isoCirc identified close to 20 times more events.

CircRNAs containing retained introns, termed as exon-intron circRNAs (EiRNAs), were previously shown to regulate gene expression in the nucleus based on a study of several such circRNAs (Nat Struct Mol Biol. 2015; 22(3):256-264). However, EiRNAs have been difficult to detect by short-read-based methods, due to the large size of human introns. Using isoCirc, we identified 720 internal alternative splicing events corresponding to EiRNAs. We illustrate a specific example (**Fig. 4c** of the revised manuscript) where the EiRNA was produced in a tissue-specific manner. Additional examples of circRNA isoforms with internal alternative splicing events and their cross-tissue patterns are shown in **Supplementary Figs. 21 and 22**. Collectively, these results expand our current knowledge about internal alternative splicing events within circRNAs. Moreover, they highlight the advantage of using isoCirc for experimental discovery of full-length circRNA isoforms, including EiRNAs.

Fig. 4. isoCirc discovery of alternative splicing events within circRNAs.

a Pie chart showing percentages of isoform pairs in which the predominant isoform (with highest median read count across 12 human tissues and HEK293 cell line for a given gene) had alternative splicing differences in BSJ only, FSJs only, or both compared to each of the other isoforms in the gene. Number of isoform pairs for each category is given in parentheses next to category name in the legend.

b Summary table showing number of internal alternative splicing events within circRNAs corresponding to four major types of alternative splicing patterns, when requiring that the minor isoform had at least 2, 5, or 10 isoCirc reads. Number of internal alternative splicing events in which the splicing events of both isoforms had FSJs annotated as FSM only or FSM/NIC are represented in two rightmost columns.

c isoCirc read coverage tracks for 12 human tissues and aggregated HEK293 replicates displaying the two most abundant circRNA isoforms of *PRPSAP1* – *PRPSAP1.circRNA.1* and *PRPSAP1.circRNA.2*, which had an alternative splicing event corresponding to a retained or spliced intron, respectively. A separate track displaying base-level conservation scores across vertebrates (phyloP 46-way) is supplied. Transcript structures and BSJs of *PRPSAP1.circRNA.1* and *PRPSAP1.circRNA.2* are shown using red boxes and black arrows. Total number of reads across all 12 human tissues and HEK293 replicates for each isoform is indicated next to the isoform identifier.

Rahimi et al. (2019) recently published a bioRxiv preprint on nanopore sequencing of full-length circRNAs ([biorxiv.org/content/10.1101/567164v1](https://www.biorxiv.org/content/10.1101/567164v1)). We are unable to make a direct comparison between our data and those of Rahimi et al., because **1**) the sequencing data of Rahimi et al. have not been publicly released, and **2**) the two studies analyzed distinct biological samples. Nonetheless, from the information provided in the Rahimi et al. preprint, we have noted several key differences between the two methods and datasets.

Firstly, unlike isoCirc which amplifies and sequences full-length circRNAs, the method by Rahimi et al. cannot be strictly considered a full-length circRNA sequencing strategy, as the circRNAs are nicked and linearized by gentle hydrolysis before nanopore sequencing. Although the authors made efforts to optimize this linearization step to prevent over-fragmentation and degradation of circRNAs, there is no guarantee that circRNA molecules are cut only once, and that the sequenced reads are from full-length circRNAs. From the numbers provided in Rahimi et al., we can infer that the majority of their nanopore reads do not map to full-length circRNAs. Specifically, in their human and mouse data, ~3.2% and ~3.3% of raw reads were mapped across circRNA BSJs (n.b. these were not necessarily full-length circRNA reads). On the basis of their circRNA enrichment, the authors assumed that reads mapping linearly inside defined circRNA regions without crossing the BSJ also originated from circRNAs. With this key assumption, the authors considered 25.1% and 26.2% of their raw reads as “circRNA mapping reads”. Extrapolating from these numbers, we can infer that at least 87.3% (i.e. $1 - 3.2\%/25.1\%$) and 87.4% (i.e. $1 - 3.3\%/26.2\%$) of such “circRNA mapping reads” in their human and mouse datasets were non-full-length, because they did not cross circRNA BSJs. Additionally, the true proportion of full-length circRNA reads among all “circRNA mapping reads” is expected to be considerably smaller than 12.7% and 12.6% (i.e. proportion of BSJ reads among all “circRNA mapping reads”), because the majority of these BSJ reads are also expected to be non-full-length.

Secondly, the Rahimi et al. dataset had a high error rate (6.8% and 6.4% per base for human and mouse respectively, as inferred from base quality scores). This high error rate is not surprising, because their method used the standard nanopore sequencing protocol to sequence each RNA only once. This high error rate made it challenging to reliably detect circRNAs, as noted by the authors. By contrast, in isoCirc, the consensus sequences called from RCA products and used in the downstream analyses had a considerably lower error rate, especially from reads with high copy numbers, as shown in the new **Supplementary Fig. 1** (see comment #7 for figure; note that the average copy number of consensus sequence per raw read was 14.5 in the HEK293 isoCirc data).

Thirdly, apart from these methodological differences, the two studies generated quite distinct datasets. Rahimi et al. analyzed one human brain sample and one mouse brain sample, from two male human donors and one male mouse, respectively. In the isoCirc study, we analyzed 12 human tissues and 1 cell line (HEK293). Each human tissue RNA was a pooled sample from tissues of multiple donors. Therefore, our dataset allows us to generate a comprehensive catalog of circRNA isoforms across diverse human tissues.

We have included a new paragraph in the discussion section of the revised manuscript, to discuss and compare the Rahimi et al. study to isoCirc. Both studies highlight the value of nanopore long-read sequencing for circRNA analysis. There are also some convergent findings; for example, both studies identified retained introns in circRNAs. We consider the two studies as complementary, and the Rahimi et al. preprint should not disqualify our work from publication. In fact, as noted above, a side-by-side comparison between these two studies reveals considerable strengths of the isoCirc method and dataset.

Online Methods:

10. Lines 206-211: *cells of what confluency were collected? Critical for circRNA research.*

Cells were collected at approximately 80% to 90% confluency. We have added this information to the Methods. Please note that this issue is only applicable to the HEK293 data, while the majority of isoCirc data generated in this work are on RNA samples from 12 human tissues.

11. Lines 213-222: *what was the reason behind using alkaline and not acidic phenol mixture for RNA?*

We apologize for this error in our text. Thank you for catching it. We used acid-phenol: chloroform: isoamyl alcohol (125:24:1, pH 4.5) for RNA extraction, and phenol: chloroform: isoamyl alcohol (25:24:1, pH 8) for DNA extraction. We have corrected the text in the revised manuscript.

12. Lines 249-2656: why ProtoScript was used and not a reverse transcriptase without RNase H activity?

As indicated on the website of the manufacturer (NEB), ProtoScript II has reduced RNase H activity. During the early stage of developing isoCirc, we tested ProtoScript II and three other RT enzymes, including Maxima H Minus (Thermo Fisher, EP0752), Superscript III (Thermo Fisher, 18080093), and M-MLV (Thermo Fisher, 28025013). We did not observe significant differences in our results when different RT enzymes were used. Therefore, we chose ProtoScript II, which was described in a patent application on the RCA reaction of circRNAs (<https://patents.google.com/patent/WO2016187583A1/>).

13. Lines 356-358: 'If a circRNA isoform had no assigned gene based on the above procedure, then it was assigned to any gene(s) having consistent strandness and at least 1 bp overlap with the circRNA isoform,...': unclear what 1 bp overlap; 'strandness' is also not very well defined throughout. Did the authors mean that their assumption was that the circRNA precursor RNA originated from a transcript overlapping with the annotated features ('genes') in either one of the directions?

The strandness of a circRNA can be defined unambiguously by its BSJ, because the consensus splice site dinucleotide motif can only appear on one strand but not the other. The great majority of circRNA isoforms were assigned to genes based on shared exons and splice sites with linear transcripts. If a circRNA isoform could not be assigned based on this procedure, we next looked for genes located on the same strand and overlapping with the circRNA of interest in their genomic coordinates (the start and end coordinates of the gene and the circRNA isoform). This is what the “at least 1bp overlap” refers to. In reality, the overlap between the circRNA and the assigned gene is typically much larger: of the 10,148 circRNA isoforms assigned to genes based on this ‘overlap’ criterion, 95.8% were completely contained within the assigned genes’ genomic coordinates. We have modified the text in the manuscript to clarify these issues, including the definition of “strandness”.

Figures:

14. Figure 2 a,b: similar comparison to the short read data is missing. As mentioned earlier, 'read count' in this case is confusing. What was the actual read coverage?

Regarding “read count”, we have clarified this issue in response to comment #7 above. The purpose of Figs. 2a and 2b is to assess the reproducibility of isoCirc among biological and technical replicates. Following the reviewer’s suggestion, we have generated a short-read version of Fig. 2a, using BSJs detected from three RNase R-treated Illumina RNA-seq libraries of HEK293 cells with ~60 million 101bp × 2 read pairs per library. This new figure is presented as **Supplementary Fig. 8** of the revised manuscript. Comparing this figure to Fig. 2a and Supplementary Fig. 3, we see that among technical replicates, isoCirc has comparable reproducibility at read count ≥ 2 and slightly higher reproducibility at read count ≥ 3, as compared to Illumina RNA-seq data on RNase R-treated samples at the same read-count threshold. This result is now described in the revised manuscript. Neither library type has >50% overlap in detected BSJs among technical replicates, indicating that the sequencing has not reached saturation for circRNA detection. We have not generated a short-read version of Fig. 2b, as short-read RNA-seq does not directly interrogate full-length circRNA isoforms, and the analysis in Fig. 2a at the BSJ-level is sufficiently informative for comparing the reproducibility of isoCirc vs short-read RNA-seq.

Supplementary Fig. 8. Pairwise comparisons of similarity between BSJs identified from three RNase R-treated Illumina RNA-seq libraries, at various read-count thresholds.

Heatmap showing pairwise comparisons of similarity between BSJs identified from three RNase R-treated Illumina RNA-seq libraries. For a library pair, the degree of similarity was calculated as the number of shared BSJs found in both libraries, divided by the total number of BSJs identified in either library. Color reflects the degree of similarity between two libraries, as indicated by the legend. BSJs with read count ≥ 2 (top) or read count ≥ 3 (bottom) were included in separate plots.

15. Figure 2c: what is the reason behind such a high variance between the replicates? Axis title is confusing, it most likely is 'Replicate', not 'Number of replicates'.

We apologize for the confusion regarding Fig. 2c, especially its x-axis. The intention of Fig. 2c is to show that the set of circRNA BSJs detected in both HEK293 biological replicates contains a higher percentage of known BSJs documented in existing circRNA databases (85.1%), as compared to the set of BSJs detected in only one of the two HEK293 biological replicates (46.9%). This result is expected. For the x-axis, "1" and "2" do not refer to replicates 1 and 2. Rather, "1" refers to circRNA BSJs detected in only 1 HEK293 biological replicate, and "2" refers to circRNA BSJs detected in 2 (both) HEK293 biological replicates. We have revised the main text and figure legend to clarify this point.

16. Figure 2e: very confusing description of 'read count'. Please clearly make a distinction between physical reads and concatemer read coverage.

We have addressed and clarified this issue in response to the reviewer's comment #7 above.

17. Figure 3: it may be OK to include these visualisations, but I do not find the presented information particularly insightful. What exonic feature differences were observed across the tissues? How do these

differences compare to the short-read inferred structure of circRNAs (solely based on back-spliced junctions and on back-spliced junctions plus isoform prevalence inference)? What are the main trends of change across the tissues? Are the changes in alternative splice sites, alternative exon inclusion, exon skipping or back-splicing at different points present?

We have conducted a detailed alternative splicing analysis of full-length circRNA isoforms identified by isoCirc. We have added a new multi-panel figure (**Fig. 4**), two supplementary figures (**Supplementary Figs. 21** and **Fig. 22**), and a new section in the main text to present these results. For a summary of these new results, please refer to our response to comment #9 above.

18. Main figures and Supplementary figures need to be re-worked to include consistent (and always readable) font size/type and have proportional elements. Currently, at an anticipated published size, nearly all important labels and many of the graphical elements will be illegible.

We appreciate the reviewer's suggestion. To ensure that main figures and supplementary figures are legible upon print, we followed guidelines for submitting figures to Nature (<https://www.nature.com/nature/for-authors/final-submission>) and edited/revised all figures in Adobe Illustrator based on these guidelines. Specifically, figures anticipated to span two-columns were rescaled to have width 183 mm and figures anticipated to span one-column were rescaled to have width 89 mm. For consistency, the range of font sizes used across all figures was 8 pt for small text objects and 12 pt for larger text objects. We also made sure that weights for lines and strokes were set between 0.25 and 1 pt to ensure legibility upon print.

Reviewer #3:

Xin et al described isoCirc, an experimental protocol and a companion computational pipeline to sequence circular RNAs (circRNA) with Oxford nanopore (ONT). While standard short-read RNA-seq can identify circRNA, the technology requires substantial sequencing depth and can't obtain full-length structure. This manuscript addresses both issues. I am not an expert on circRNAs or experimental protocols. I will focus on the computation and analysis part of this manuscript.

We thank the reviewer for the positive comment about the technological innovations of this work. We have revised the manuscript based on the reviewer's requests and suggestions, as detailed below.

1) I can access the UCSC custom track provided by the authors. However, their code repository seems private. I am unable to access it. The circRNAs the authors found have been deposited to GEO, but the data are also private. I would appreciate if the authors can upload the source code and the circRNAs in HEK293 as supplementary data (or put them on a department FTP server). This will help me to evaluate the results.

In the original submission, the GEO dataset was private. We shared a token for reviewer access, but apparently this mechanism did not allow the reviewers to access the companion raw isoCirc data deposited into SRA. There is no good mechanism to share access to a private Github repository. Although we uploaded the entire software package as a Supplementary File to our manuscript, both reviewers 1 and 3 missed this file. To address these issues, we have now made both the Github repository and GEO/SRA data public, so the reviewers should have no problem accessing the software and data.

2) A key question I have is: how many circRNAs the authors found are real and how many are transcription noises? Fig S1/S2 suggest that two replicates are more consistent if we increase the count threshold to 3. Fig 2c shows the majority of circRNAs found in one replicate only are "novel". Are these really novel? Can authors find a way to estimate the fraction of real circRNAs?

The reviewer asked an important question regarding the reliability of the identified circRNA isoforms, and how many of them are nonfunctional (i.e., "transcription noises"). In the context of transcriptomics, and, more specifically, the discovery of alternative transcript isoforms, "real" and "functional" isoforms are two distinct concepts. Many "real" transcript isoforms (i.e., *bona fide* transcripts produced in cells) arise from splicing noise and are nonfunctional (PLoS Genet. 2010; 6(12):e1001236). Like all sequencing strategies for transcript isoform discovery, isoCirc allows us to discover novel circRNA isoforms, but it does not directly yield information about which isoforms are functional. Identification of functional circRNAs is a major challenge for the community and is beyond the scope of this work. Nonetheless, by generating and sharing the most comprehensive dataset of full-length circRNA isoforms ever produced for human transcriptomes across many tissue types, our work provides a highly valuable resource that will facilitate and motivate future functional studies of circRNAs in diverse biological systems.

On the other hand, a major goal of this work is to identify "real" circRNA isoforms by minimizing noise and artifacts in isoCirc data. This aspect is particularly important, given the high error rate of nanopore long reads. We achieved this goal by using a combination of experimental and computational procedures. Experimentally, we generated rolling circle amplification (RCA) products of circRNA isoforms. RCA products contain multiple copies of a template circRNA sequence. Consensus sequences called from RCA products are expected to have a reduced error rate, thus improving downstream analyses. To confirm this, for all isoCirc reads for which consensus sequences were called and mappable, we estimated the error rates of raw reads and of consensus sequences called from raw reads with different copy numbers. Raw reads had the highest error rate. As expected, we observed a copy number-dependent decrease in the error rate of consensus sequences, with consensus sequences called from raw reads with >10 copies having the

lowest error rate. We include this information in the new **Supplementary Fig. 1** of the revised manuscript:

Supplementary Fig. 1. Comparison of error rates for raw isoCirc reads and called consensus sequences across six HEK293 libraries.

Violin plots showing distributions of error rates for raw isoCirc reads and for consensus sequences called from 2, 3, 4, 5, 6 to 10, and more than 10 copies of a template circRNA sequence across six HEK293 libraries. isoCirc reads that had consensus sequences called and were mappable to the human genome are represented in the plots.

Computationally, we used stringent criteria to remove potential false-positive detection of circRNA BSJs and full-length circRNA isoforms. All circRNA BSJs reported by isoCirc are “high-confidence” BSJs, meaning that they must pass stringent filters to remove consensus sequences with low mapping quality and spurious alignments. For BSJs involving novel splice sites to be reported, the filtering criteria are much stricter (as compared to BSJs involving known splice sites), requiring near-perfect alignment between the isoCirc consensus sequence and the corresponding genomic sequence around the BSJ. This aspect was described in the Supplementary Methods. We now make it explicit in the main manuscript as well. Furthermore, we required a read-count threshold of 2 in a given tissue (a common threshold used in the RNA-seq literature; e.g., see *Nucleic Acids Res.* 2010 38(14):4570-8) to select circRNA BSJs and full-length isoforms for downstream analyses. Users of our data can choose a more stringent read-count threshold (3 or higher) to select circRNAs.

Here are a few of my thoughts, though I am not sure if they work.

a) Replot Fig. 2a, 2b, 2c, 2d, 3a, Fig S1 and S2 for read count threshold 1 through 4. This will give readers an idea about whether the results are sensitive to the count threshold. If the results are sensitive to the threshold, the authors will probably get more isoforms with deeper sequencing. Is that right?

For RNA-seq, we expect to detect more isoforms with deeper sequencing until the sequencing depth reaches saturation. Due to the relatively low yield and high per-sequence cost of long-read sequencing (vs short-read sequencing), almost none of the long-read RNA-seq studies of human transcriptomes published to date are sufficiently deep to reach saturation. We expect this to be the case for our isoCirc work as well. The read count threshold will affect the number of splicing events and isoforms detected, as would be the case for almost all RNA-seq studies.

Based on the reviewer's suggestion, all of the plots referenced above have been updated to include a read-count threshold of 1 through 3. A read-count threshold of 2 is the conventional threshold in the RNA-seq literature to select reliable events (e.g., see *Nucleic Acids Res.* 2010 38(14):4570-8). Thus, we believe that the range of 1 to 3 appropriately enables readers to observe how our results are affected by the threshold. Because this manuscript already has a large number of figures and supplementary figures (4+23), we chose not to include plots for read-count threshold of 4, to avoid making our data presentation and figures overly complex.

We have added a note in the discussion section, stating that we do not expect isoCirc at the current sequencing depth to have reached saturation, and that more circRNA isoforms will be detected with deeper sequencing. Nonetheless, the current isoCirc dataset already represents the most comprehensive dataset of full-length circRNA isoforms ever produced for human transcriptomes, and will be a highly valuable resource for the research community.

b) The authors are looking for canonical splice signals around back-splicing junctions (BSJs) within a window. Is it harder (e.g. more distant to alignment ends) to find splice signals around less confident and novel BSJs?

The procedure to look for canonical splice-site signals around alignment ends, followed by local re-alignments to assess alignment quality, is a common practice, dating back 20 years to computational detection of exon-intron structures and alternative splicing events from expressed sequence tags (*Nucleic Acids Res.* 2001;29(13):2850-2859). Furthermore, all BSJs reported by isoCirc are "high-confidence" BSJs, meaning that they must pass stringent filters to remove consensus sequences with low mapping quality and spurious alignments. For BSJs involving novel splice sites to be reported, the filtering criteria are much stricter (see response to comment #2 above for full details).

c) Some (not all) BSJs are formed due to inverted repeats in flanking introns. Have the authors checked this feature? How many BSJs have inverted repeats nearby? Is the fraction of inverted repeats reduced among novel BSJs? The general idea is to stratify a biologically meaningful feature by the confidence level of BSJs. The authors have done that in Fig S1 and S2 but I think they should do more to understand the quality of their data.

We appreciate the reviewer's suggestion. We have performed a new analysis mimicking the analysis conducted by Dr. Li Yang and colleagues on complementary sequence-mediated exon circularization (*Cell* 2014;159(1):134-147). Specifically, for each high-confidence BSJ identified in HEK293 cells, we checked a certain window of the genomic sequence flanking the back-splice sites. We asked whether this window contained inverted Alu repeats in the convergent orientation, divergent orientation, or both orientations, or did not contain any inverted Alu repeats. We analyzed two window sizes (1000 and 2000 bp). We analyzed four sets of BSJs (known or novel BSJs detected in both replicates or only one replicate), plus a fifth set of negative-control BSJs by making 10,000 random pairs of a downstream 5' splice site and an upstream 3' splice site, using non-BSJ splice sites based on isoCirc results and circRNA databases. As illustrated in the new **Supplementary Fig. 12**, a higher proportion of known BSJs had inverted Alu repeats in their flanking regions as compared to novel BSJs, and a higher proportion of BSJs detected in both replicates had inverted Alu repeats as compared to BSJs detected in only one replicate.

Both known and novel BSJs were significantly enriched for inverted Alu repeats as compared to the negative-control BSJs. We have included new text and a new **Supplementary Fig. 12** to describe this result:

Supplementary Fig. 12. Fractions of known or novel BSJs identified by isoCirc with inverted Alu repeats in flanking introns.

Stacked barplots showing fractions of known or novel BSJs identified by isoCirc with inverted Alu repeats in flanking introns with window sizes 1,000 bp (top) and 2,000 bp (bottom). Inverted Alu repeats were classified based on their orientation as convergent (red), divergent (blue), both (green), or no inverted Alu repeats detected (purple). BSJs with read count ≥ 2 (left column) and read count ≥ 3 (right column) were classified based on their annotation status and level of detection across biological replicates of HEK293 cells, as follows: BSJ is novel and detected in only one replicate (1Novel); BSJ is novel and detected in both replicates (2Novel); BSJ is known and detected in only one replicate (1Known); and BSJ is known and detected in both replicates (2Known). A fifth set of negative-control BSJs (Negative) was constructed by making 10,000 random pairs of a downstream 5' splice site and an upstream 3' splice site, using non-BSJ splice sites based on isoCirc results and circRNA databases (circBase and MiOncoCirc).

3) If a circRNA doesn't have a forward-splicing junction (FSJ), i.e. it consists of a single exon, will it be counted as full splice match (FSM)? Perhaps that is a new category for FSJ?

The definition of FSM/NIC/NNC was adopted from long-read RNA-seq analysis of linear RNA transcripts in the SQANTI software (Genome Res. 2018;28(3): 396-411). All single-exon circRNAs (without FSJ) were considered to be FSM for the FSJ classification. This aspect was noted in Methods.

Minor comments:

4) Supplementary Methods is not long and does not contain figures. I think it can be moved to Online Methods.

Supplementary Methods describes the details of the isoCirc computational pipeline to identify high-confidence BSJs and full-length circRNA isoforms. This document is useful for readers to fully

understand the isoCirc computational pipeline. In particular, it describes the steps taken to ensure reliable results, and the stringent criteria applied to remove potential false-positive detection of circRNA BSJs and full-length circRNA isoforms due to isoCirc reads with low mapping quality and spurious alignments. However, this document has too much technical information for inclusion in Methods. We appreciate the reviewer's suggestion but prefer to keep this document as Supplementary Methods.

5) In Fig. 1a/1b, Fig S1 and S2, please write the exact "Degree of similarity" numbers in the heatmap. There should be enough space to put these numbers in the plots. It is difficult for me to tell what the similarity is by looking at the colors.

We thank the reviewer for this suggestion, which was also made by reviewer 1. We have added the actual value for the degree of similarity among a given pair of libraries (i.e., number of BSJs or full-length isoforms detected in both libraries, divided by total number of unique BSJs or full-length isoforms detected in either library) to Figs. 2a and 2b, as well as Supplementary Figs. 3 and 4. As illustrated in these figures, the degree of similarity among replicates was higher when we required that identified BSJs (Supplementary Fig. 3) or full-length isoforms (Supplementary Fig. 4) be supported by a larger number of isoCirc reads. As expected, technical replicates from the same biological replicate had slightly higher similarity than technical replicates from different biological replicates.

REVIEWERS' COMMENTS

Reviewer #1 (Remarks to the Author):

The authors did an excellent job to address my comments, and I have no further comments.

Reviewer #2 (Remarks to the Author):

The authors have thoroughly addressed the suggestions raised during the peer review and have substantially improved the manuscript.

I believe the manuscript now meets the expectations of a publication in Nature Communications and can be published upon minor language/typo checks/edits.

A comment to the authors' response "We are not entirely sure about the context of the 50x coverage required by ONT, as noted by the reviewer, but we guess that this refers to a requirement regarding ONT-based genome sequencing. In transcriptomics, we typically do not see this kind of metric used to describe sequencing coverage, because the transcriptome is not uniform. Here, we used read count ≥ 2 as the threshold for the number of independent nanopore reads supporting a given transcript. This is the conventional threshold used in the RNA-seq literature to select reliable events (e.g., see Nucleic Acids Res. 2010 38(14):4570-8). We also present data based on a more stringent threshold (read count ≥ 3)." is listed below.

In transcriptomics this metric has not been widely adopted for the exact same reason that reads from short read sequencing cannot always be unanimously attributed to a transcript (iso)type, while most of the read identification and mapping relies on the pre-existing genomic (transcriptomic) reference. Although, coverage and average coverage are still often used. With the long read sequencing, it is possible to unanimously attribute reads to a transcript (iso)type in most cases, apart from cases with very small-scale differences, such as editing, SNPs etc. Thus, coverage criteria to precisely meet the requirements of de novo/reference-free reconstruction of the transcripts' sequences would be similar to those applied for the genomic DNA. However, I think in the rolling circle amplification approach used by the authors, this criteria is eased with the copies of the same sequence encountered in each read, which arguably can only carry only a very low additional intrinsic error rate characteristic to the polymerase precision. Thus, in this case, the effective coverage would equal average of [read counts attributable to the same transcript (iso)type] X [number of sequence repeats in each read] X [sequence repeat unit length-adjusted polymerase precision], which I thought could be useful to additionally highlight in the manuscript.

Dr Nikolay Shirokikh

Reviewer #3 (Remarks to the Author):

The authors have addressed most of my major concerns, in particular with the added supplementary Figure S12. I have a new minor comment related to this figure.

When describing the figure in the main text, the authors did a Fisher exact test (line 202) comparing the observation against the negative control. This test is non-interesting as it is expected to be significant. The reasoning is that even with the most lenient thresholds, there are a fraction of functional circRNAs. Given large numbers, a Fisher exact test will almost certainly lead to a very significant P-value.

Here is my interpretation of Figure S12. The figure implies that regardless of the thresholds in use, known circRNAs exhibit similar features in terms of inverted ALUs, which is encouraging. Given that

18% ($=1-8166/(8166+252+785+797)$) of negative controls have inverted ALUs within 2kb, if we *assume* 56% ($=1-6287/(6287+1250+3543+3327)$) of functional circRNAs have inverted ALUs, we can infer that only 26% of "1Novel" observations (or 55% of "2Novel") are functional by solving a linear equation. This suggests that a great fraction of novel circRNAs are either bioinformatics errors/transcription noises or look very different from known circRNAs, which is fine as the authors have probably found thousands of functional circRNAs in the novel category. This is what I meant by "to stratify a biologically meaningful feature by the confidence level of BSJs" in the first round of review. I hope the authors may properly discuss Figure S12. A Fisher exact test is not informative.

Detailed Responses to Reviewer Comments

Reviewer #2

The authors have thoroughly addressed the suggestions raised during the peer review and have substantially improved the manuscript. I believe the manuscript now meets the expectations of a publication in Nature Communications and can be published upon minor language/typo checks/edits.

A comment to the authors' response "We are not entirely sure about the context of the 50x coverage required by ONT, as noted by the reviewer, but we guess that this refers to a requirement regarding ONT-based genome sequencing. In transcriptomics, we typically do not see this kind of metric used to describe sequencing coverage, because the transcriptome is not uniform. Here, we used read count ≥ 2 as the threshold for the number of independent nanopore reads supporting a given transcript. This is the conventional threshold used in the RNA-seq literature to select reliable events (e.g., see Nucleic Acids Res. 2010 38(14):4570-8). We also present data based on a more stringent threshold (read count ≥ 3)." is listed below.

In transcriptomics this metric has not been widely adopted for the exact same reason that reads from short read sequencing cannot always be unanimously attributed to a transcript (iso)type, while most of the read identification and mapping relies on the pre-existing genomic (transcriptomic) reference. Although, coverage and average coverage are still often used. With the long read sequencing, it is possible to unanimously attribute reads to a transcript (iso)type in most cases, apart from cases with very small-scale differences, such as editing, SNPs etc. Thus, coverage criteria to precisely meet the requirements of de novo/reference-free reconstruction of the transcripts' sequences would be similar to those applied for the genomic DNA. However, I think in the rolling circle amplification approach used by the authors, this criteria is eased with the copies of the same sequence encountered in each read, which arguably can only carry only a very low additional intrinsic error rate characteristic to the polymerase precision. Thus, in this case, the effective coverage would equal average of [read counts attributable to the same transcript (iso)type] X [number of sequence repeats in each read] X [sequence repeat unit length-adjusted polymerase precision], which I thought could be useful to additionally highlight in the manuscript.

We thank the reviewer for recommending the manuscript for publication after the previous round of revision. As for the reviewer's comment regarding a measure of transcriptome coverage, while we appreciate the reviewer's comment and discussion, we believe this is a quite peripheral and minor issue not essential to our work, and we prefer to leave this out of our manuscript.

Reviewer #3

The authors have addressed most of my major concerns, in particular with the added supplementary Figure S12. I have a new minor comment related to this figure.

When describing the figure in the main text, the authors did a Fisher exact test (line 202) comparing the observation against the negative control. This test is non-interesting as it is expected to be significant. The reasoning is that even with the most lenient thresholds, there are a fraction of functional circRNAs. Given large numbers, a Fisher exact test will almost certainly lead to a very significant P-value.

*Here is my interpretation of Figure S12. The figure implies that regardless of the thresholds in use, known circRNAs exhibit similar features in terms of inverted ALUs, which is encouraging. Given that 18% ($=1-8166/(8166+252+785+797)$) of negative controls have inverted ALUs within 2kb, if we *assume* 56% ($=1-6287/(6287+1250+3543+3327)$) of functional circRNAs have inverted ALUs, we can infer that only 26% of "1Novel" observations (or 55% of "2Novel") are functional by solving a linear equation. This suggests that a great fraction of novel circRNAs are either bioinformatics errors/transcription noises or look very different from known circRNAs, which is fine as the authors have probably found thousands of functional circRNAs in the novel category. This is what I meant by "to stratify a biologically meaningful feature by the confidence level of BSJs" in the first round of review. I hope the authors may properly discuss Figure S12. A Fisher exact test is not informative.*

We thank the reviewer for the positive comment about our revision. We agree with the reviewer's comment about Figure S12, and we have revised our text on Page 10 of the manuscript based on the reviewer's recommendation:

“Both known and novel BSJs (versus negative-control BSJs) were significantly enriched for inverted Alu repeats (p value < 3.6e-50 in all comparisons, Fisher’s exact test). As expected, a higher proportion of known BSJs (versus novel BSJs) had inverted Alu repeats in their flanking introns, and a higher proportion of BSJs identified in both biological replicates (versus BSJs in only one replicate) had inverted Alu repeats (Supplementary Fig. 12). Therefore, a higher proportion of novel BSJs lacked the characteristic feature of inverted Alu repeats in flanking introns for their biogenesis.”